# Asynchronous evolution of interdependent nest characters across the avian phylogeny

Yi-Ting Fang[1], Mao-Ning Tuanmu [2] & Chih-Ming Hung [2]

Nest building is a widespread behavior among birds that reflects their adaptation to the environment and evolutionary history. However, it remains unclear how nests evolve and how their evolution relates to the bird phylogeny. Here, by examining the evolution of three nest characters—structure, site, and attachment—across all bird families, we reveal that nest characters did not change synchronically across the avian phylogeny but had disparate evolutionary trajectories. Nest structure shows stronger phylogenetic signal than nest site, while nest attachment has little variation. Nevertheless, the three characters evolved interdependently. For example, the ability of birds to explore new nest sites might depend on the emergence of novel nest structure and/or attachment. Our results also reveal labile nest characters in passerines compared with other birds. This study provides important insights into avian nest evolution and suggests potential associations between nest diversification and the adaptive radiations that generated modern bird lineages.

---

[1] Department of Life Sciences, National Chung Hsing University, Taichung, Taiwan. [2] Biodiversity Research Center, Academia Sinica, Taipei, Taiwan. Correspondence and requests for materials should be addressed to M.-N.T. (email: mntuanmu@gate.sinica.edu.tw) or to C.-M.H. (email: cmhung@gate.sinica.edu.tw)

Almost all birds build nests, ranging from a simple scratch on the ground to complex woven structures hanging in trees[1]. Since nests are important for determining reproductive success, they can be critical to avian evolution[2]. Dedicated nests that protect eggs and chicks from external disturbances might be the central reason that birds survived through the Cretaceous–Tertiary (K–T) boundary (65 Ma), during which non-avian dinosaurs went extinct[1] (but see ref. [3]). In addition, nest characters are adaptively associated with the environments in which birds breed[2] and could be subject to sexual selection[4]. Thus, nest differentiation might mirror the diversification of birds that breed in different habitats and sexual selection could reinforce the differentiation[4,5]. Surprisingly, the "big picture" of avian nest evolution remains unclear, mainly due to the lack of systematic nest characterizations across modern birds[6,7]. The availability of nearly complete avian phylogenies[8–10] and ample records of bird nest characters (e.g., the Handbook of Birds of the World Alive (HBW))[11] make it possible to examine nest evolution on a broad scale.

Nest building is likely a complex of multiple behavioral characters. For example, birds need to select their nest sites first, and then build their nests with particular structures and attach the nests to supporting objects. Studies have implied that different characters of avian nests, such as structure and site, are evolutionarily interdependent[12–15], but empirical evidence is restricted to individual species or small groups of species. A more extensive investigation is needed to examine whether avian nest characters change synchronously and thus can be treated as a character "syndrome" or individual nest characters have independent evolutionary histories and thus should be examined separately.

Birds of the same taxonomic groups are often observed to build similar nests[6]. For example, almost all pigeon species build flimsy, shallow nests[11]. Therefore, nest characters can be expected to reflect avian phylogenetic history[16]. By contrast, given that avian nesting behavior is sensitive to climatic and environmental changes[17], ecological adaptation and convergent evolution may occur so frequently that they obscure phylogenetic signals[5]. Although studies have found phylogenetic signals for nest characters in a few avian families[18,19] (but see ref. [20]), it is still unclear whether the signal is prevalent across all modern bird families for most characters due to a lack of comprehensive studies and statistically-based evidence.

To better understand the evolution of nests across the avian phylogeny and its role in avian evolution, we characterized the nests of all bird families in three dimensions: site, structure, and attachment. We examined the evolutionary history of these three characters by reconstructing their ancestral states along phylogenetic hypotheses for all avian families. Using conventional and evolutionary model-based Mantel tests (see Methods), we revealed different levels of phylogenetic signal in the three characters. In spite of their disparate evolutionary patterns, our results showed that the three nest characters evolved interdependently. In addition, different evolutionary patterns of nests found in passerine families, which represent approximately one half of all bird families, further suggested that a better understanding of avian nest evolution requires a comprehensive investigation across the entire avian phylogeny.

## Results

**Evolution of nest characters.** We coded nest site, structure and attachment for all 242 avian families (Supplementary Data 1, 2) based on the descriptions of nest characteristics in the HBW. Among the seven types of nest structures (see Methods for details), cup nests were most common across modern bird families, followed by secondary cavities, domed and platform nests (Fig. 1a). The distribution of those nest structures was uneven between passerines (Passeriformes) and non-passerines (Fig. 2a). Most families building cup and domed nests were passerines, whereas almost all families building platform nests were non-passerines (Fig. 2a). Most non-passerine cavity nesters were landbird families and closely related to one another (e.g., families belonging to the orders Strigiformes, Leptosomiformes, Trogoniformes, Bucerotiformes, Coraciiformes, and Piciformes), whereas passerine cavity nesters were mainly distinctive families (Fig. 2a). About one-fifth of avian families make scrape nests, all of which were non-passerine families (Figs. 1a and 2a).

While most (ca. 2/3) avian families nest in trees, >20% of the families nest in non-tree vegetation (including bushes, bamboo, and thick tangled herbaceous vegetation), on the ground or on cliffs/river banks (Fig. 1b). Each of the remaining three nest site types was used by <10% of avian families (Fig. 1b). Nest site distributions were also different between passerines and non-passerines. Most ground-nesting families were non-passerines, whereas most of the families nesting in non-tree vegetation were passerines (Fig. 2b).

Among the four types of nest attachment, basal attachment was most common, with 80% of all families using only this approach and 90% using this approach along with others (Fig. 1c). Each of the other three attachment approaches (i.e., lateral, horizontally

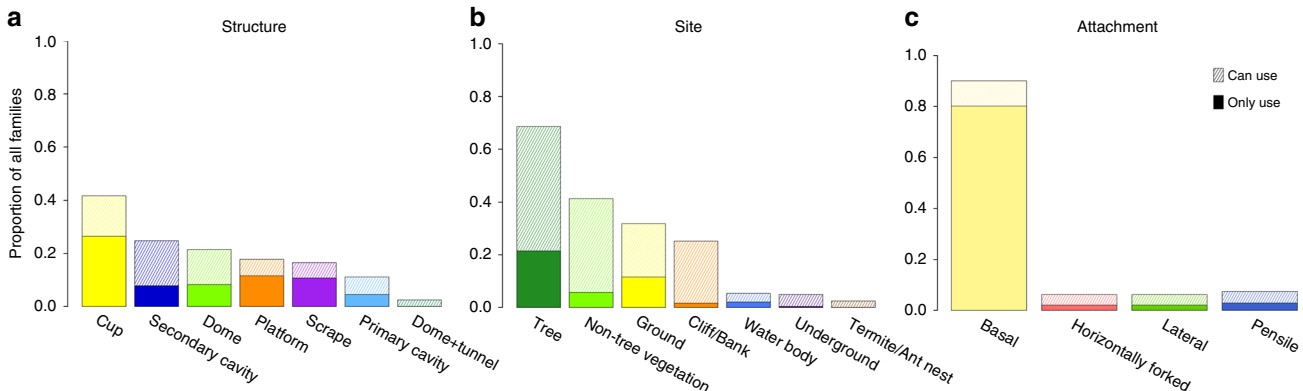

**Fig. 1** Frequency of nest character states distributed across avian families. Proportion of avian families using the character states of **a** nest structure, **b** nest site, and **c** nest attachment. A filled bar indicates the proportion of avian families whose nests only have the focal character state, whereas a striped bar indicates the proportion of avian families whose nests have mixed states including the focal state

forked and pensile attachment) was used by <10% of all families and almost all of which were passerines (Figs. 1c and 2c).

To examine the evolutionary trajectories of the three nest characters, we used BayesTraits[21–23] to reconstruct ancestral

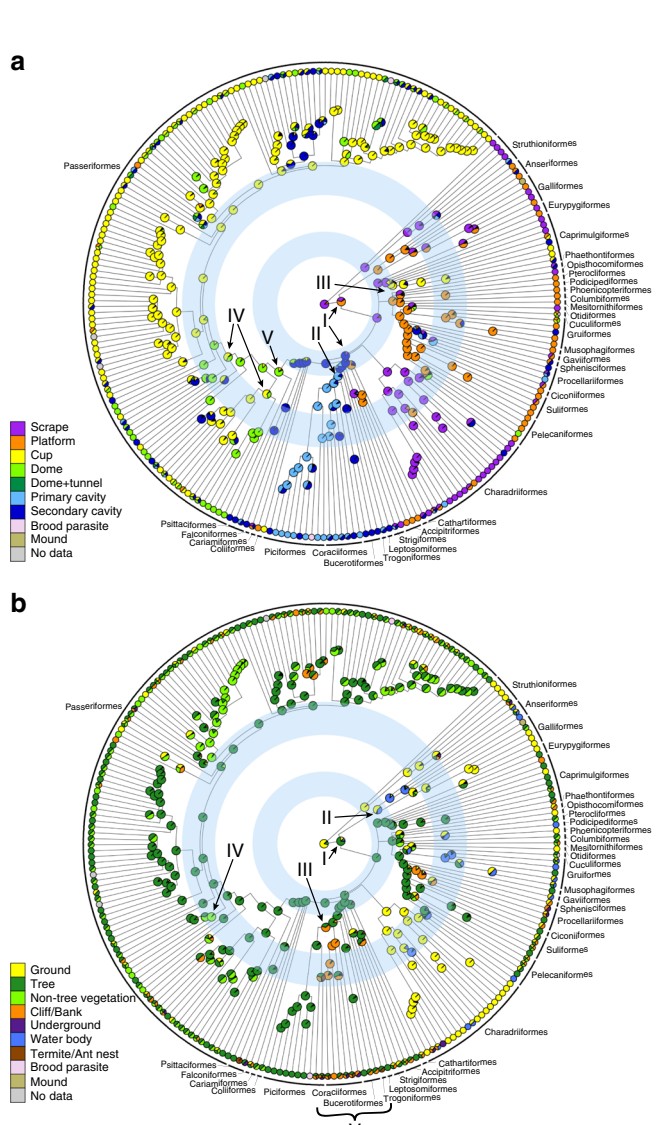

**a**

**b**

**c**

character states for each character across the avian family phylogeny. To account for phylogenetic uncertainty, the ancestral state reconstruction was based on 1000 avian family trees obtained from BirdTree.org[7,24]. This analysis showed that the common ancestor of all modern birds most likely made scrape nests on the ground (Fig. 2a–c). In the evolution of nest structure, platform nests and secondary cavity nests were derived from scrape nests early in avian evolutionary history (I in Fig. 2a). The primary cavity nests evolved after the secondary ones (II in Fig. 2a). Cup nests evolved independently at least three times during avian evolutionary history. One time was before passerines arose and was restricted to the lineage containing swifts, treeswifts and hummingbirds (belonging to Caprimulgiformes, III in Fig. 2a), and the other two times were early in passerine evolution, making it a prevalent character state in the order (IV in Fig. 2a).

For the evolution of nest sites, results showed that tree nesters evolved relatively early (I in Fig. 2b), and those nesting on water bodies, cliffs/banks or in non-tree vegetation (II, III or IV in Fig. 2b, respectively) appeared later in the evolution of birds. Underground nests occasionally and independently evolved across the avian phylogeny. Birds nesting in termite/ant nests mainly belonged to Coraciimorphae, including kingfishers (Coraciiformes), rollers (Coraciiformes), trogons (Trogoniformes), jacamars (Piciformes), puffbirds (Piciformes), etc. (V in Fig. 2b). Nest attachment approaches were highly conserved (Fig. 2c). Approaches other than basal attachment (e.g., I, II, and III in Fig. 2c) evolved relatively late in avian evolutionary history and evolved independently several times in modern and ancestral passerines, except for the lineages including swifts and treeswifts (Caprimulgiformes; IV in Fig. 2c).

The ancestral state reconstruction revealed several intriguing patterns suggesting interdependence in the evolution of the three characters; the results were further supported by coevolution analyses conducted using BayesTraits. For example, only after platform or cavity nests evolved from scrape nests (I in Fig. 2a), did birds start nesting on other nest sites, including trees, water bodies and cliffs/banks (I, II and III in Fig. 2b, respectively). A coevolution analysis between nest structure (scrape vs. non-scrape) and nest site (ground vs. non-ground) showed that the two characters were interdependent (Bayes factor = 52.3 for a dependent model). Estimated transition rates between character states suggested a higher probability that the evolution of non-ground structure was driven by that of non-scrape nests than the other way around (Supplementary Fig. 1a).

Similarly, the use of non-tree vegetation (IV in Fig. 2b) as nest sites occurred after the appearance of cup nests in the evolutionary history (IV in Fig. 2a). The coevolution analyses showed that the evolution of the cup structure and non-tree

**Fig. 2** Phylogenetic distribution of nest character states. Nest character state distributions and ancestral state reconstructions of **a** nest structure, **b** nest site, and **c** nest attachment across an avian family phylogenetic tree. The filled colors of circles at tips and nodes of the tree represent nest character states for modern families and their ancestors, respectively. A circle filled with multiple colors indicates a family or an ancestral taxon with more than one character state. The families that are brood-parasitic, build mound nests, or have no nest data were excluded from analyses as their nests cannot be characterized, although they are shown in the figures for the purpose of comprehension. The blue rings indicate two major adaptive radiations in modern bird evolution. The Roman numerals indicate events discussed in the Results. The avian order names are labeled on the tree (see Supplementary Fig. 8 for family names labeled on the tree)

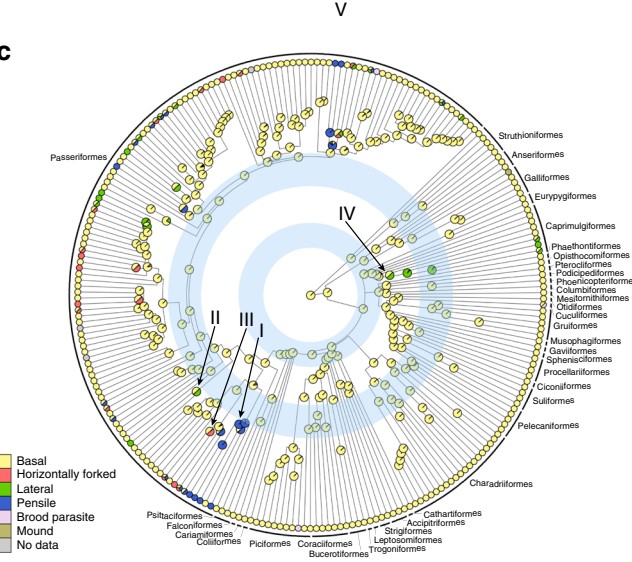

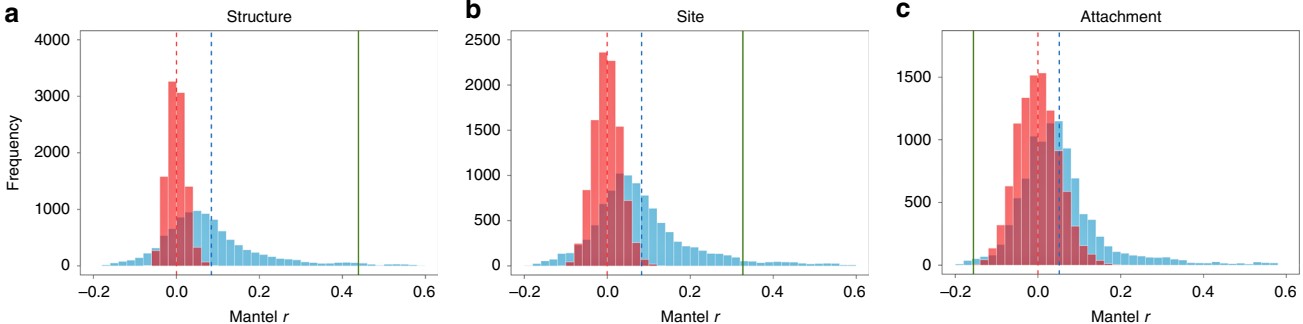

**Fig. 3** Phylogenetic signal tests for nest characters. Mantel coefficients under the null hypotheses in conventional Mantel (red histograms) and EM-Mantel (blue histograms) tests. **a** Nest structure, **b** nest site, and **c** nest attachment. Red and blue dashed lines indicate the means of the expected values in conventional Mantel and EM-Mantel tests, respectively. Green lines indicate the empirical values for the three nest characters

vegetation sites was not independent (Bayes factor = 35.7 for a dependent model). The use of the cup nest structure was more likely to drive than to be driven by the use of non-tree vegetation (Supplementary Fig. 1b). In addition, the coevolution between the non-basal attachment and domed/cup structure was also strongly supported (Bayes factor = 34.8 for a dependent model). The non-basal attachment approaches (e.g., I, II, III, and IV in Fig. 2c) evolved only in the lineages using domed or cup nests (e.g., III, IV, or V in Fig. 2a), and the evolution of the cup/domed nest structure was more likely to drive than to be driven by that of the non-basal attachment (Supplementary Fig. 1c).

**Phylogenetic signal of nest characters**. To examine the phylogenetic conservatism of the three nest characters, we assessed the level of phylogenetic signal for each of the characters using both conventional and evolutionary model-based Mantel (EM-Mantel) tests[25]. While the conventional Mantel test examines whether more closely related families have more similar character states (i.e., a positive Mantel correlation), the EM-Mantel test further contrasts the empirical phylogenetic signal to that simulated from a neutral evolution model (a null model)[25]. The results of the Mantel tests showed different levels of phylogenetic signal for the three nest characters. Nest structure showed a significantly positive correlation between nest character dissimilarity and phylogenetic distances among all avian families ($P < 0.01$) and the positive correlation was significantly larger than the expectations based on a neutral evolution model ($P = 0.02$), suggesting strong phylogenetic conservatism (Fig. 3a; Supplementary Table 1). Thus, there was a stronger resemblance between nest structures from birds of closely related families than suggested by the evolutionary distances. Although nest site showed a significantly positive Mantel correlation ($P < 0.01$), the positive correlation showed no or only marginally significant differences from neutral model simulations ($P = 0.05$; Fig. 3b; Supplementary Table 1). Surprisingly, nest attachment showed a significantly negative Mantel correlation ($P < 0.01$), which was also significantly different from the neutral model simulations ($P = 0.01$; Fig. 3c; Supplementary Table 1). The unusual correlation might be caused by high resistance to change in nest attachment (see Discussion for details).

The analyses based solely on passerine families showed different patterns. Although nest structure ($P < 0.01$) and attachment ($P = 0.01$) showed significantly positive Mantel correlations, nest site ($P = 0.43$) did not exhibit a Mantel correlation significantly different from zero (Supplementary Table 1). In addition, the EM-Mantel tests for all three characters did not show any significant signal of phylogenetic conservatism

(nest structure: $P = 0.13$; nest attachment: $P = 0.13$; nest site: $P = 0.31$; Supplementary Table 1), suggesting labile nest evolution in passerine families.

## Discussion

The divergent evolutionary patterns among nest site, structure and attachment found in this study suggest that bird nests contain a set of distinct characters, which may have been subject to different selective forces. Thus, bird nest characters should not be treated as a character "syndrome" because they do not change synchronically across the avian phylogeny. This implies that studies focusing on a single nest character may only capture parts of the complex interactions among nest characters. In addition, different evolutionary patterns of the nest characters between passerines and all bird families further illustrate that evolutionary inferences depend on the phylogenetic levels. Therefore, to properly understand the evolution of avian nests, the nest characters should be examined under a research framework that considers multiple characters as separate but interdependent components and spans a full avian phylogenetic spectrum.

The EM–Mantel tests revealed a stronger phylogenetic signal (i.e., greater phylogenetic conservatism)[26,27] in nest structure than site. This suggests that changes in nest structure probably require changes in complex construction skills[28] and thus are subject to strong genetic constraint. By contrast, the changes in nest sites might be less genetically constrained or more frequently affected by external factors, such as intra- or inter-specific competition or nest predation[29,30]. For example, Coraciimorphae birds (i.e., trogons (Trogoniformes), kingfishers (Coraciiformes), cuckoo rollers (Leptosomiformes), woodpeckers (Piciformes), etc.) nest exclusively in cavities, but the cavities can be in trees, ant/termite nests, on cliffs/river banks, or underground (Fig. 2a, b). In addition, larger intraspecific variation in nest site choice than in nest structure has also been found for several bird species[15].

Interestingly, the Mantel test results indicated that avian families' nest attachment approaches might be more similar to those of distantly related families than more closely related ones. This relationship is likely due to a highly conserved evolution of nest attachment approaches. The basal attachment is the most common approach and the others have evolved only within passerines, with an exception of one non-passerine lineage including hummingbirds, swifts and treeswifts (Caprimulgiformes, Fig. 2c). There are many cases in which one passerine family shares the basal attachment with non-passerine families but not with its sister family or closely related passerine families, and this leads to the negative correlation. This argument is

further supported by the evidence that neither conventional Mantel nor EM–Mantel tests showed a negative relationship for nest attachment among passerine families (Supplementary Table 1). A highly conserved character can result from strong benefits of a common state, functional interdependence with other characters, and/or insufficient genetic variation[26]. The basal attachment appears to be the easiest and most efficient way to provide support for nests against gravity under most conditions, and it can be used to attach nests with different structures to diverse types of sites. Non-basal attachment approaches are mainly restricted in passerines probably because they are more effective when being used with domed or cup nests than other structure types. Our coevolution tests support the evolutionary dependence of non-basal attachment on domed or cup nests. Finally, it is also possible that non-basal attachment approaches evolve more slowly due to a lack of genetic variation in nest attachment-related genes.

The multifaceted evolution of bird nests revealed in this study sheds light on the long-term debate over their phylogenetic conservatism. Several studies have found congruent patterns between phylogeny and nest characters in some families[5,18,19]. By contrast, other studies suggest that nest characters are too labile to reflect phylogeny[20]. The discrepant conclusions can be caused by (1) investigation of different nest characters, (2) different ways of interpreting the analytic results or (3) idiosyncratic nature of nest evolution among different lineages. First, a previous study focusing on the nest structure of the swallow family (Hirundinidae) shows a strong phylogenetic signal in their nests[18]. However, when we reanalyzed their data and focused on nest site, the congruent level between the nest character and phylogeny became lower (Supplementary Fig. 2). The reanalyzed results are consistent with our conclusion based on data across all avian families that changes in nest site are more labile than changes in nest structure.

Second, the criteria to determine phylogenetic conservatism might largely affect the conclusions of the previous studies. If phylogenetic conservatism is claimed when any character shows phylogenetic signal (i.e., a soft criterion), multi-character studies are likely to conclude conserved evolution in nests. By contrast, if phylogenetic conservatism is claimed only when all studied characters show strong phylogenetic signals (i.e., a hard criterion), multi-character studies tend to indicate labile evolution in nests. For example, a study of swiftlet (Aerodramus) nests suggests no phylogenetic signal because all characters showed non-significant signal except one[20]. However, if the authors had followed a soft criterion, they would have concluded that swiftlet nests show detectable phylogenetic conservatism.

Finally, different families may show different levels of phylogenetic signals in their nest characters. This argument implies that studies focusing on a single or subset of families could present an incomplete view of avian nest evolution. For example, our study shows that the phylogenetic signals in nest characters among passerine families are weaker than those among all avian families as a whole, suggesting that the passerines' nest characters might have experienced more frequent convergence or local adaptation[31]. Although weak genetic constraint can also lead to low levels of phylogenetic signal, this is less likely to be the reason. That is because compared with non-passerines, passerines tend to build nests with more elaborate nest structures (i.e., domes or cups) and attachment approaches (i.e., lateral, horizontally forked and pensile attachment). More manipulative movements are needed to complete the elaborate nests, which are found to be associated with higher levels of cerebellar foliation[28] and thus are more likely to require the aid of new genetic variation than relaxation from genetic constraint[32]. In addition, although relatively high levels of taxonomic and phylogenetic uncertainty

among passerine families could also explain their low levels of phylogenetic signals, this should not be the main reason. That is because we compared the phylogenetic signals of two datasets both containing the same passerine families, and thus the biased effect caused by phylogenetic uncertainty in passerine families should be canceled out or largely reduced. Instead, the large number of passerine species that have repeatedly adapted to diverse habitats in a relatively short period of time[33] might better explain the seemingly labile pattern for all of their nest characters[31]. Overall, evidence for the multifaceted nature of the avian nesting behavior across a large avian phylogeny can help resolve the debate over phylogenetic conservatism in bird nest evolution.

Considering the multidimensional but interdependent nature of avian nest characters, we provide a novel view on avian nest evolution and its association with the evolution of modern birds. It has been argued that the diverse modern bird lineages resulted from two large-scale adaptive radiations[33–37] (indicated by the two blue rings in Fig. 2). The first one, resulting in the major avian lineages (around or above the level of order)[34], might have been facilitated by empty ecological niches left by the extinction of non-avian dinosaurs. Results of this study showed that the first adaptive radiation coincided with the appearance of diverse nest structures derived from the scrape nest (Fig. 2a), which resembles the nest evolution in non-avian dinosaurs but reaches to more complex structures[3]. The nest structure diversification might, in turn, have allowed birds to explore new nest sites and fill opened niche spaces. For example, after non-passerine birds evolved to build platform and cavity nests, they were ecologically released to explore nest sites other than the ground, such as trees, water bodies, cliffs/banks, the underground and termite/ant nests (Fig. 2a, c). Thus, the derived types of nest structure might be involved in the diversification of birds as novel organismal features that are argued to contribute to adaptive radiation[5,38].

More unique organismal features in passerines are likely to have aided in the second adaptive radiation by occupying novel niches[38–41]. As most plants went extinct at the K–T boundary, vegetation nest sites might not have been fully available until the recovery of plant diversity in the Eocene[42], which coincided with the rapid diversification of passerines[33]. Thus, the newly evolved (or recovered) plant species might have provided ecological opportunities with new nest sites to passerines. With compact nest structures (i.e., small and tightly interwoven cups and domes) that allow flexible placement of nests in vegetation, passerines intensified their use of trees as nest sites and further explored lower vegetation layers (e.g., bush and strong herbaceous vegetation). Furthermore, unique nest attachment approaches that evolved relatively late can be another key feature further promoting passerines' evolutionary success (Fig. 2c). Lateral, horizontally forked and pensile attachments may have allowed passerines to attach their cup and domed nests to new locations on trees (e.g., canopy edges or terminal braches) and non-tree vegetation (e.g., vines or reeds) to advance the utilization of refined nest niches.

It is noteworthy that although our study agrees that the domed nests evolved earlier than cup nests in passerines[41], the cup nests of passerines could be a result of "reverse evolution". Cup nests have appeared in non-passerine lineages, particularly the one containing swifts, treeswifts and hummingbirds (Caprimulgiformes; III in Fig. 2a), before passerines evolved. Thus, the cup nests of passerines might have resulted from evolutionary restoration to a former state via using ancestral genetic variation. However, it is also possible that the passerine cup nests resulted from convergent evolution based on a different genetic mechanism from non-passerine ones.

Although the novel traits (e.g., nest structure and attachment) and ecological opportunities (e.g., nest site) discussed above

provide a reasonable explanation for avian adaptive radiation[38], it is still a hypothesis. We hope that the hypothetical links among the diversification of nest characters, ecological niches and avian lineages highlighted in this study will stimulate more in-depth examinations. For example, these hypothetical relationships can be tested by comparing nest character changes among avian lineages that have experienced various levels of differentiation rates and ecological niche dynamics at different evolutionary time points. It is also worth pointing out that nest site, structure and attachment are not the only facets of avian nesting behavior. Future studies should also consider other characters, such as nest materials and nest construction procedures.

## Methods

**Nest characters**. We obtained the information regarding nest structure, site and attachment for all 242 avian families from the HBW (data accessed in October 2017; see below for details). We defined the states of nest structure by modifying the nest descriptions of Neotropical birds in a previous study[7] and categorized them into seven types: scrape, platform, cup, simple dome, dome with tunnel, primary cavity and secondary cavity. The scrape type indicated that eggs were laid in a place with no obvious nest-construction or with only brief scratching or cleaning. A platform nest was a nest where feathers, leaves, sticks or vines were stacked or loosely intertwined to form a platform. A cup nest can be distinguished from a platform nest by an erected, surrounding rim made by interweaving nest materials or mud. Cup nests could vary in their depth but parental birds could not hide their whole bodies inside the nests (i.e., expose whole or partial bodies outside), whereas domed nests referred to the nests where parental birds could sit inside without exposing any part of their bodies. We further divided domed nests into those with a tunnel exit and those without because the tunnel part presumably requires more complex construction skills. Cavity nests were divided into primary cavities, which were excavated directly by the birds using them, and secondary cavities, which were generated naturally or originally excavated by other animals. Although sometimes nests with different structures (such as platform, cup or domed) were built inside cavities, they were categorized as cavity nests in this study (following ref. [7]), because such nests shared the same advantages and disadvantages of being located inside cavities with simple cavity nests, and information regarding nest structure inside cavities was not always available.

We defined nest sites as the locations where nests were built and categorized them into seven states: ground, tree, non-tree vegetation, cliff/bank, underground, water bodies, and termite/ant nests. A ground site was recorded when birds laid eggs on the ground with or without building nests, and a tree site was recorded if a nest was mentioned by the HBW to be in trees. The non-tree vegetation sites included those on bushes, bamboo or thick tangled herbaceous vegetation (such as vines or reeds) that occupies forest understory, grassland or wetland habitats. When birds built nests in cliffs, river banks, or piles of soil or rocks, the sites of these nests were categorized as cliff/bank. For the nests built in burrows underground or in termite/ant nests, they were assigned as underground and termite/ant nest, respectively. If the nests were built on the surface of water or piled up from the bottom of lakes or ponds and thus surrounded by water, their sites were assigned as water bodies.

We also defined the states of nest attachment by modifying the descriptions of Neotropical birds in ref. [7] and categorized them into four approaches: basal, lateral, horizontally forked and pensile attachment. The basally attached nests referred to those supported mainly from their bottom, including those sitting among multiple interweaving tree or bush branches and those inside cavities. The laterally attached nests were those attached to supporting objects such as branches (other than horizontally forked branches, see below) or rocks solely by their lateral parts. The horizontally forked attachment was the approach by which nests were attached to two or more horizontally forked tree or bush branches by their lateral parts (e.g., drongos' nests). The horizontally forked attachment presented a unique approach compared with other lateral attachment types. The pensile attachment referred to the approach that nests were hanged down from a supporting object with its upper, narrower attachment part.

We separated birds that exhibit brood parasitism into a different category from the ones above because we could not characterize their nests. We also separated birds in the megapode family (Megapodiidae) into a different category because they do not construct nests but build mounds to lay their eggs inside and utilize solar or fermentation heat to incubate eggs (although some megapode species build burrows, we considered the burrows essentially as mounds because the parents also do not incubate their eggs and the burrows resemble the early stage of mound building). We treated the character states of those families as missing data in further analyses.

For each family, we recorded all states mentioned in the family summaries in the HBW for every nest character ("all" states). When there were two or more states of any character recorded for a family, we checked nest information at the species level to determine if >10% of the species in that family used a particular state. We considered all states used by more than 10% of the species as "effective" states. For the families in which fewer than 50 species had nest information

available in the HBW, we used the information from as many species as possible. For those containing >50 species with nest information available, we used the information from 50 species approximately evenly distributed across all genera in that family and estimated the percentage of each nest character state. The reason for using a threshold of 50 species was that the value is close to the mean number of species in each bird family (10,978 species/242 families = 45). For the families with no or out-of-date family summaries when we accessed the HBW (i.e., 48 new or recently revised passerine families in the October 2017 version of HBW), we recorded their "all" and "effective" character states based solely on the species level information. To assess the potential effects of the arbitrary percentage threshold on our conclusions, we repeated the same analyses with the datasets of "all" and "effective" states. Since the results based on the two datasets were very similar, we mainly reported the results based on the "effective" dataset in the main text and mentioned those based on the "all" dataset in supplementary information (Supplementary Figs. 3–5; Supplementary Table 2).

**Nest character evolution based on family phylogenetic trees**. We randomly extracted 1000 avian family phylogenetic trees containing all 242 families from http://birdtree.org based on the Hackett constraint[8,24] by selecting one species from each family as a representative. In general, we selected one species from the type genus (which defines the family and the root of the family name) of the focal family as the representative (222/242 families). For example, we selected *Sitta europaea* as the representative of the family Sittidae. To reconstruct ancestral character states for each of the three nest characters (i.e., structure, site and attachment), we used BayesTraits v.2.0[21–23] with the 1,000 family trees being used to take into account the uncertainty in estimates of trees and their branch lengths. Each run of the Markov chain Monte Carlo (MCMC) algorithm contained 20 million iterations following a burn-in of 2 million iterations; sampling was done in every 10,000 iterations. The acceptance rates between MCMC iterations were auto-adjusted to 20 – 40% to improve convergence of the results (according to the BayesTraits manual). We ran three chains for each analysis to assess convergence of the results by checking their MCMC trend lines (see Supplementary Fig. 6 for the trend line assessment). The reported parameters were averaged across the three chains. We modeled the transition rates between different character states as a single value with an exponential prior with a mean of 0.1. We also built models with different transition rates for different pairs of character states. However, model comparisons based on Bayes factors indicated better performance of the equal-rate model in nest site and attachment, and the parameter estimates in the different-rate model for nest structure could not converge even after 40 million iterations or with different priors (Supplementary Fig. 7). Thus, we only reported the results based on the equal-rate models. We generated a majority-rule consensus tree from the extracted 1000 trees using the R package, phytools[43] with branch length estimated using a "least square" approach (Supplementary Fig. 8). We mapped the states of each nest character on the consensus tree to summarize and visualize the distribution of current and ancestral character states.

**Evolutionary interdependence between nest characters**. We used the "Discrete" function in BayesTraits[44] to examine evolutionary interdependence between nest characters. Given that the "Discrete" function could only be applied to binary characters, we converted the original multi-state characters to binary ones for this analysis. We examined evolutionary interdependence and transition directions among character states for three specific cases. First, we examined whether birds started nesting on sites other than ground after the emergence of structure types other than scrape, and re-coded families as either scrape or non-scrape for nest structure and either ground or non-ground for nest site. Second, we examined whether non-tree vegetation was used as a nest site after birds built cup nests by re-coding families as cup or non-cup structure and non-tree vegetation or other site types. Finally, we examined whether the non-basal attachment approaches evolved after the lineages started building domed or cup nests by re-coding families as either basal or non-basal attachment and either cup/domed or other structure types. If a family contained multiple character states, we recoded the family as the later derived state, based on the results of ancestral state reconstruction (Fig. 2). For example, if a family contained both species with scrape and non-scrape nests, we recoded the family as a non-scrape nester. We tested evolutionary interdependence between characters by comparing a 'dependent' model, in which the transition rates between states of one character depend on those of the other character, with an 'independent' model, in which the two characters evolve independently[44]. The transition rates in each model were estimated based on the 1,000 family trees using the MCMC algorithm with an exponential prior with a mean of 0.1. We ran three separate analyses for each model, with each containing a MCMC chain of 200 million iterations and a burn-in of 20 million iterations. We sampled the parameter estimates every 10,000 iterations and reported the averages of the three analyses. We used Tracer v1.6 (http://beast.bio.ed.ac.uk/Tracer) to ensure adequate mixing and properly high effective sample sizes (ESS > 500) for posterior probability distributions. We used Bayes factors calculated from the harmonic means of log likelihoods to compare models, with a value larger than 5 indicating strong support for the 'dependent' model[23].

We also used the MuSSE (Multi-State Speciation and Extinction) function of the diversitree[45] package in R to test the effect of nest structure types on the

speciation rates of avian lineages across the bird phylogeny and the detailed methods and results were shown in Supplementary Note 1.

**Phylogenetic signal tests**. Mantel tests were used to measure phylogenetic signals for nest characters. The conventional Mantel test contrasted the matrices of empirical correlations between phylogenetic distance and trait dissimilarity to that obtained from a tree with traits randomly permuted without taking into account any evolutionary model. By contrast, an evolutionary model-based (EM-Mantel) test contrasted the empirical correlation to the one obtained from a tree with traits mapped by a neutral evolutionary model[25]. Therefore, by explicitly incorporating an evolutionary model, the EM-Mantel test provided a more realistic null hypothesis to test if correlation deviated from the model expectation and assess the level of phylogenetic signal. We modified the original R code provided by ref.[25] for the EM-Mantel test to make it applicable to multiple character states within families using the Bray-Curtis dissimilarity statistic to measure state dissimilarity between families. We built the evolutionary model by using the rTraitDisc function in the ape package[46] in R with the state transition rates and the root state being set as the values estimated by the BayesTraits analyses. We randomly permuted the character states and ran the neutral evolutionary model 9,999 times to obtain the null distributions of the Mantel coefficient (r) for the conventional Mantel and EM-Mantel tests, respectively. To test for potential biases caused by analyses based on a subset of data, we performed the above analyses for both all families and solely passerine families.

**Code availability**. The computing code used in this study is available in the Figshare database (https://doi.org/10.6084/m9.figshare.6126641)[47].

**Data availability**. The data that support the findings of this study are available in the supplementary information files and the Figshare database (https://doi.org/10.6084/m9.figshare.6126641)[47].

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

# Acknowledgements

We thank Bob Zink, John Wang, and Noah Last for comments on the manuscript. We are grateful to Chun-Cheng Huang for helping collect data and improving the manuscript. The research was initiated with the support from the summer internship program at Biodiversity Research Center, Academia Sinica. We also acknowledge financial support from Academia Sinica.

## Author contributions

C.-M.H. and M.-N.T. conceptualized the study. Y.-T.F. collected the data. All the authors contributed to data analyses and manuscript writing.

## Additional information

**Competing interests:** The authors declare no competing interests.

