## [Peer Review File · Nature Communications]

Reviewers' comments:

Reviewer #1 (Remarks to the Author):

I think the issue addressed in this manuscript is one that is timely and this analysis could provide a very important contribution to our understanding of the causes of diversity in nest structure, location and attachment type. I have, however, a number of questions ranging from the quite substantial to the relatively trivial, which I think need addressing in some form before the value of this work will be obvious to Nature Communication readers. Generally, I think the authors need to work much harder on being clear about the rationale for the work and when interpreting the value of their findings. Neither of these are yet sufficient at this point, in my view, but I hope that by addressing my comments, both of these will become both clearer and, thereby, more compelling.

L 27-28 Not sure what this means: novel nest structure, facilitated by advanced attachment approaches, might allow birds to explore new nest sites

L 28-33: this needs to convey a clearer message: Nevertheless, more labile evolution of all nest characters found among passerine families suggests inconsistent patterns between different levels of phylogenetic coverage. These results not only provide the first thorough view on avian nest evolution, but also reveal the potential association between nest diversification and adaptive radiations that generated modern bird lineages

L44: If the nest fails, either by placement or structure, reproductive success is likely to be nil, so this statement could be much stronger

L48: Not all birds living in the same/similar habitat build the same nest structure, the same locations, or use the same materials, only some evidence for this but needs a more substantial analysis: nest characters are strongly tied to the environment where birds live

L56: Not clear what is meant here, given the references: Studies have implied the interdependence of different components in avian nesting behavior¹¹⁻¹⁴

L59-61: not sure this can be concluded conclusively from a phylogenetic analysis, especially one at family level

L62: no evidence for this statement and no need for it. There is increasing evidence (since the 2008 Walsh et al. paper) for a role for learning and memory in a variety of aspects of building and there is a lot of data that show flexibility of site use following loss of a nest to predation/poor weather conditions (Lima 2009).

L86: why choose the family as the level of analysis? For example, the Hall et al. (2015) data show considerable variation within a single family.

L169: the authors haven't analysed bird nesting behavior, they've analysed nest structure and location, neither of which are behaviours. They should change this to bird nests

L173: it would be helpful to be clearer what is meant here, without the reader having to read the papers referenced and discover what the authors of the current ms are implying here. Also see lines 175-179: what do the authors really mean here?

L183: do the known within-family data support this claim?

L192-193: could the authors say this more clearly and just as a biological statement: Interestingly, we found a negative relationship between the dissimilarity in nest attachment approaches and the phylogenetic distance across all bird families.

L209: lack of genetic variation in what?

L229: what is 'complex'?

L245: 'to build', rather than 'to use'; is it the structure that has evolved or the choice of material used, which might strongly affect the structure that can be built or the choice of location, which might then impact on the structure?

L256: what is sophisticated about a cup or a dome (see the Price and Griffith paper for domes preceding cups)? Do they require a larger number of manipulative movements in their production? A greater variety of materials?

L276: 'intentional'?

L277: does this exclude mud cups?

L282: why make this assumption? What is it based on (see point re L256 above)

L287: this then means that you decided on nest structure based on its function (i.e. the value or not of being inside a cavity), not on the building required. Did you actually use this decision rule for all of your classifications? Also see L321 with regard to classification of variation within the Megapodiidae.

L290: a very pedantic point but a nest is built and material deposited

L296: how did you determine the strength of the plants involved?

L297: what is understory relative to bushes?

L324: what data did you survey from the HBW? This suggests that you looked at species but analysed the data at the family level. Is this what you did? Its not really very clear how you dealt with multiple states within a family, especially if there was more than two.

This manuscript is really missing some good illustrations of the variation the authors are seeking to explain.

Reviewer #2 (Remarks to the Author):

Until recently global analyses of avian nest evolution were handicapped by the unavailability of a complete avian phylogeny, lack of information on nests for almost a third of the world species (especially tropical), and the difficulty of accessing the existing nest data scattered through primary, often obscure, sources. All this has changed over the last decade with the unveiling of a comprehensive avian tree (Jetz et al), publication of the HBW series (where nest data are summarized for all behaviorally known species), and the avalanche of new nest data coming from the New World tropics. The authors of this manuscript, and Price & Griffith (P&G) earlier this year, are the first to reap the benefits of these new advances and data availability. Whereas P&G analyzed evolution of nest architecture across all passerines (60% of all birds of the world), the authors of this study are the first to tackle all birds of the world (at the family level). Their concept and methodology are original and should be of interest to ornithologists and broader audience interested in animal construction and evolution of behavior.

The authors should be commended for recognizing that avian nesting behavior is multifaceted and should not be reduced to merely one or two parameters, as was done in most prior studies. The three parameters selected in this study (site, structure, and attachment) are indeed among the most fundamental and informative. The first two correspond to 'location' and 'shape' in the study of P&G. Type of nest 'attachment' is the third parameter which, in my opinion, makes a better and more independent character than 'exposure' selected by P&G. However, the authors should explicitly acknowledge that site, structure, and attachment are not the only facets of nesting behavior to consider. Future studies should include such parameters as type of material, construction method, nest ontogeny (i.e., sequence of behavioral steps taken in nest construction), etc.

In the Discussion, the authors state that "previous studies that failed to recognize the multifaceted nature of avian nesting behavior are likely to have captured only parts of the complex interactions among nest characters and thus reached inconsistent conclusions on the evolutionary trajectory of avian nests" (lines 171-174) and that treating nesting behavior as a single character "has prevented them from capturing the whole picture of avian nest evolution" (lines 211-214). Although true in the sense that no study to date, including this contribution, could capture the full complexity of avian nest evolution, these claims are overstated and unfounded. I can't think of any "inconsistent conclusions" in the four studies cited (refs 4, 18, 19, 20) and I don't see how the only example given, the recent shift in some swallows "from nesting in natural habitats to human buildings" (lines 214-219), is relevant to any of these studies. As an author of one of these studies (ref. 19, Zyskowski & Prum 1999 [not 1990, as cited]), I object to listing it as an example of "studies that failed to recognize the multifaceted nature of avian nesting behavior" (line 174). In fact, this study is based on 24 different nest characters!

Critical to this study are the details of character coding and scoring across avian taxa. The

authors did a good job identifying broad categories of nest site, structure, and attachment. Similarly, for this rather coarse analysis, it was reasonable to: (1) treat taxa with unique forms of nesting (brood parasites, mound-builders, no nest built) as missing data; (2) code all cavity-nesters regardless of nest form as just 'cavity'; and (3) implement 10% threshold for coding polymorphisms. However, I find it problematic that avian families as defined in HBW (source of nest data) are not all the same as family-level clades in Jetz et al (source of phylogenetic trees). Here are some examples from Tables S1 & S2 of spurious coding resulting from HBW's outdated family definitions:

- Paradisaeidae: coded 'dome' because this family in HBW still includes Cnemophilidae
- Sylviidae: coded 'dome' because this family in HBW still includes Cettiidae, Phylloscopidae, etc.
- Turdidae: coded 'dome' because this family in HBW still includes Brachypteryx
- Eupetidae coded 'dome' because this family in HBW still includes Lesser Melampitta
- Thraupidae: coded 'dome' because this family in HBW still includes Euphonia, Chlorophonia
- Fringillidae: coded as only 'cup' because this family in HBW excludes genera Euphonia & Chlorophonia

Most families in Tables S1 and S2 are listed in phylogenetic order but there is some intermixing of passerine and non-passerine families. Since only orders are listed in the figures, it would be helpful to add a column for order into both tables and assign the families accordingly.

Tables S1 & S2 also include a few errors and omissions:

- Procellariidae: should not have 'dome' and 'dome+tunnel' in both tables
- Cathartidae: should include 'secondary cavity' in both tables
- Furnariidae: should include 'dome+tunnel' in both tables
- Tyrannidae: should include 'pensile' attachment in both tables
- Emberizidae: should include 'dome' in both tables
- Corvidae: should include 'secondary cavity' at least in Table S1
- Pomatostomidae: attachment listed as 'pensile' in both tables, but only 1 species (Garritornis isidorei) builds pensile nests; change to 'base' in Table S1 and to 'base; pensile' in Table S2
- Meliphagidae: attachment listed as 'pensile' in both tables, but only 2 species of Ramsayornis build pensile nests, thus 'pensile' should be excluded from Table S1

Although most of these details are relatively minor and probably wouldn't change the overall conclusions of the paper, it would be prudent to confirm this prior to publication. I find the conclusions of the paper plausible. The manuscript is well written and organized and the figures informative and quite handsome.

Kristof Zyskowski

Reviewer #3 (Remarks to the Author):

This study looks at the evolution of several aspects of avian nests (structure, support, site) across all avian families using a published phylogeny. Although exploration of avian nest characteristics have been conducted within several families, this has not been previously done at this level. The results allow some exploration of which characters are more labile, the order of evolution of some states, and the timing of these events allowing some comparison across the different traits. I think the overall questions are worth addressing, and some interesting conclusions were found. Others are not unexpected, however, and I think there might be more insights that could be gained (making this of greater significance to the field). Additionally, I feel there are some limitations in the methods (detailed below) that I feel limit some of the strength of the results.

First, the introduction emphasized that the availability of complete phylogenies made it possible to examine nest evolution at this scale (line 51). However, the analyses were then restricted to the family level in spite the availability of a complete (all species) phylogeny. Since many characters were polymorphic within families (and the authors had to modify the code used for one test to accommodate the polymorphism), I am not clear why the analyses were not done at the species level. The exclusion of some character states (those found in less than 10% of species) would not be an issue if all species were examined, and it would prevent rare character states from being "over-weighted". Additionally, while being over-weighted is one possibility for rare character states, there are cases where a rare character state may be biologically important to consider. For example, if an early diverging species/clade within a family exhibits a character state not found in later diverging species, it may be the ancestral condition. By excluding it in the analyses, you miss this possibility, whereas coding species would allow inclusion of this trait AND the pattern of trait changes within a family (as determined using the branching patterns of the species in the family) should allow more rigorous assessment of the ancestral states of a family. It is not clear why this more rigorous analysis was not conducted, given that it should have been possible.

Second, although phylogenetic uncertainty was accounted for in some ways, by downloading 1000 trees, these were then used to make a single consensus tree to use for character state reconstruction. The more rigorous approach would be to analyze each of the 1000 trees (BayesTraits should allow this to be done), and then use those results to assess reconstructions and look for patterns. Obviously this is more challenging (summarizing the 1000 sets of results), but it also can provide much greater information about the confidence of character state reconstruction at a node. For example, in a consensus tree, some clades may be the "best" that can be found, but still may have shown up in only a very small percentage of trees (I have seen consensus trees where a clade was only found in 2 or 3 % of all trees). When a consensus is used, this node is considered to be equal in confidence to a node that was found in 100% of all trees, even though that poorly supported node is not even found in a majority of the trees. This is most likely to affect reconstruction of traits during the rapid radiations (nodes in the gray shaded regions of figure 2), and that is where several of the focal nodes (labeled with letters) are located, suggesting some conclusions

may be based on reconstructions that may have low confidence.

Specific comments:

Line 30: I was unclear what phylogenetic coverage meant, but I presume this means level (among orders versus within a single order?).

Line 51: Two of the three citations given to represent "complete phylogenies" are not very complete, and instead include only 0.5% or 2% of all avian taxa. Only Jetz et al. is complete. There are other complete, or more complete phylogenies that are published for birds that would be more appropriate to cite here (Holt et al. 2013, Burleigh et al. 2015) if a desire is to show that trees encompassing a large amount of avian diversity are available.

Line 69: I might rephrase this as "across all modern bird families", given that this is not looking at patterns within families (unlike the studies cited in line 68), but instead at a coarser level.

Line 116-117: What is the confidence about the assertion that secondary cavity nesters evolved first? This occurs during the rapid radiation, and it appears that primary cavity nesting shows up very shortly after secondary. This is an example where a more rigorous treatment of the 1000 trees might yield some different conclusions (or reduce confidence in the conclusion).

Line 117-118: Is it possible to actually estimate the number of gains of cup nests (using a pre-defined criterion)? The arrow D points to two nodes, suggesting that 3 is most likely (one outside of passerines, and two noted here). If the evidence is more equivocal, perhaps give a range (3-10 times). That would make it easier to evaluate how labile this trait may be for readers.

Lines 132-139: This raises some other interesting questions. What about loss/reversibility (e.g., are there some character states do not appear to be reversible, when a character state changes is it more likely to revert to an earlier state or go to a novel state)? The focus on what is presented is what might precede other things, but there are other insights that might come from this analysis that I encourage authors to explore.

Line 148: In the preceding paragraph, there is some directionality that is pointed out (what states precede other states). While not all traits may have directionality, or within a trait not all states may work in a directional way, it is clear there is some directionality (and it is easy to see how some states are likely to have evolved from other states, leading to directionality in some cases). Therefore, doing an analysis assuming non-directionality may not be a good fit to the data. Can the appropriateness of this assumption be explored or tested to ensure that these conclusions are not biased by assumption violations?

Line 173-174: Given that the cited studies were restricted to within families, where the range of states may be lower (and the power to assess patterns may be lower), could that also explain the inconsistencies that are referred to?

Lines 182-183: I am not clear why nest structure would exhibit both greater directional selection and greater conservation, at least if I understand the meaning of what is being said here. It would seem that these are opposite (if there is high directional selection, then it would suggest low conservation).

Line 224-226: Could errors in the passerine tree be an alternative explanation as well? Oscine passerine families are still not completely well understood, but it is known that traditional classification of families (which underlies parts of the Jetz et al. phylogeny) are often quite different from families circumscribed using molecular data.

Lines 304-315: How were cavity nests coded for attachment site?

Lines 325-328: I applaud coding polymorphisms by providing the different character states, and not selecting the most common state to represent a family. However, if the number of species within a family exhibiting each state is not known, how can it be known whether 10% of species exhibited a particular character state or not? Much greater clarification of what this means and how it was calculated should be provided.

Line 339: I think it may be clearer to just indicate you used the trees based on the Hackett constraint (for those not familiar with the bird tree, just indicating Hackett constraint in parentheses may be confusing).

Line 339: The Hackett et al. tree differs from the Jarvis and Prum trees in several ways, not just Coliidae (please note spelling) - and even the Jarvis and Prum trees differ from each other. Therefore, I do not see why Coliidae was excluded, yet other likely conflicting placements were retained. Possibly the other conflicting placements were not included in the constraints used by Jetz et al. (some well-supported relationships in Hackett et al. were not included in the constraint tree), but if so please state you carefully checked all Jetz et al. relationships against Jarvis and Prum, and excluded all families that were in conflict to ensure the same rules for inclusion in the tree were applied equally across the tree.

Line 343-344: Was the consensus tree made in phytools? (given how it is written, it is not clear). More importantly, was the method used to generate the consensus one that maintained branch length information (and made consensus branch lengths) or not (most consensus methods only retain relationships, and ignore branch lengths). This should be mentioned explicitly as branch length treatments can have profound effects on the results.

Lines 347-349: Was convergence assessed in some way? This is important to demonstrate.

Another point to consider might be to try and look more rigorously at the potential interdependence of states. As stated in the discussion (e.g., line 179), the traits are interdependent to at least some degree. For example, it is hard to see how a non-basal attachment could be functional for a scrape nest. However, these analyses focused on a single trait at a time without any assessment of any interdependence (except in comparing results of analyses of one trait with results of analyses on another trait). It might provide a

more robust picture of how nests evolve over time to try and look at traits together to explore the interdependence (though I admit, off the top of my head I am not certain the best approach for doing this, other than doing an evolutionary PCA in phytools).

The discussion suggests the role for nest structures as leading to adaptive radiations in birds. While this may be true, there are not explicit analyses to address this, nor are any verbal arguments of alternative hypotheses considered. Thus, while I think this is worth retaining, it should be clear it is one possible explanation and I might reduce a little emphasis on this point.

Responses to Reviewers' Comments

We thank the reviewers for their time and effort in helping us improve our manuscript. The comments and suggestions provided by the reviewers are constructive and helpful.

Our answers for each comment/question are shown after the symbol ">>" and in *italic*.

Reviewers' comments:

Reviewer #1 (Remarks to the Author):

I think the issue addressed in this manuscript is one that is timely and this analysis could provide a very important contribution to our understanding of the causes of diversity in nest structure, location and attachment type. I have, however, a number of questions ranging from the quite substantial to the relatively trivial, which I think need addressing in some form before the value of this work will be obvious to Nature Communication readers. Generally, I think the authors need to work much harder on being clear about the rationale for the work and when interpreting the value of their findings. Neither of these are yet sufficient at this point, in my view, but I hope that by addressing my comments, both of these will become both clearer and, thereby, more compelling.

L 27-28 Not sure what this means: novel nest structure, facilitated by advanced attachment approaches, might allow birds to explore new nest sites

>>*To clarify this sentence, we rewrote it and added an example (L27-29). We think that the revision makes this sentence clearer.*

L 28-33: this needs to convey a clearer message: Nevertheless, more labile evolution of all nest characters found among passerine families suggests inconsistent patterns between different levels of phylogenetic coverage. These results not only provide the first thorough view on avian nest evolution, but also reveal the potential association between nest diversification and adaptive radiations that generated modern bird lineages

>> *We agree that the sentences in our original manuscript were not very clear. So we rewrote the sentences (L29-32) and hoped that the revision makes the messages to be conveyed clearly enough under the constraint of word limits.*

L44: If the nest fails, either by placement or structure, reproductive success is likely to be nil, so this statement could be much stronger

>> *We thank the reviewer for this good suggestion. To make the statement stronger, we rewrote this sentence as "Since nests are important in determining reproductive success of birds, they can be critical to avian evolution" (L38-39).*

L48: Not all birds living in the same/similar habitat build the same nest structure,

the same locations, or use the same materials, only some evidence for this but needs a more substantial analysis: nest characters are strongly tied to the environment where birds live

>> *We agree with the reviewer that not all bird species living in the same habitat have the same nest characters. The point that we wanted to make in the original sentence was that birds' nest characters are adaptive results associated with the habitats where they breed. Changes in nest characters may either contribute to or result from differentiation between bird lineages that breed in different habitats. In addition, such differentiation may be even reinforced by sexual selection because some evidence suggests that avian nesting behavior may be subject to sexual selection. To clarify this, we changed the sentences to "nest characters are adaptively associated with the environment where birds breed and could be subject to sexual selection." and " Thus, the differentiation of nests might mirror the diversification of birds breeding in different habitats, and could be reinforced by sexual selection" (L42-45).*

L56: Not clear what is meant here, given the references: Studies have implied the interdependence of different components in avian nesting behavior¹¹⁻¹⁴

>> *We rewrote this sentence as "Studies have implied evolutionary interdependence between different characters of avian nests, such as nest structures and nest sites" (L53-55) to make the point clearer.*

L59-61: not sure this can be concluded conclusively from a phylogenetic analysis, especially one at family level

>> *In fact, phylogenetic analyses are one of the best approaches to test the temporal order of character evolution. Mapping different characters on the phylogenetic tree allows an examination of synchronicity in the evolution among different characters. Whether a family-level phylogenetic analysis is appropriate depends on the characters that are examined and the scale of interest. Since the focus of this study is on the general evolutionary patterns across the entire avian tree, we believe that variations in the nest characters among avian families, especially accounting for within-family heterogeneity, can effectively reveal the broad evolutionary patterns. In addition, a family-level analysis also has less bias caused by phylogenetic uncertainty than do a species-level analysis (please see more detailed explanations for conducting a family-level analysis below).*

L62: no evidence for this statement and no need for it. There is increasing evidence (since the 2008 Walsh et al. paper) for a role for learning and memory in a variety of aspects of building and there is a lot of data that show flexibility of site use following loss of a nest to predation/poor weather conditions (Lima 2009).

>> *To avoid the controversy over the role of learning in bird nesting behavior (which is not the focus of this study), we deleted the statement "without much learning".*

L86: why choose the family as the level of analysis? For example, the Hall et al. (2015) data show considerable variation within a single family.

>> *It is true that some avian families, especially those contain a large number of species (like Timaliidae, the family studied by Hall et al. 2015) have considerable*

within-family variation in nest characters. Actually, the within-family variation can be clearly seen in Figure 2 in our manuscript. However, instead of conducting our analyses at the genus or species level, we did them at the family level but took into account within-family trait variation in the analyses mainly for three reasons. The first one is data completeness. The information of nest characters is not available for every species. By contrast, Handbook of the Birds of the World provides explicit descriptions of nest characters at the family level, which allow us to characterize and compare nest characters across all bird families. Second, the phylogenetic and taxonomic uncertainties considerably increase below the family level, which could make genus- or species-level analyses not robust. Finally, with variation within the analytic unit ("family" in our study), we can better measure trait dissimilarity among the units in the Mantel tests based on categorical trait data. The within-unit variation allowed the dissimilarity to be measured at a nearly continuous scale. Therefore, due to the above reasons, we believe that focusing on family-level analyses can provide more robust and meaningful results. Similarly, Price & Griffith (2017) have also suggested that nest characters are generally conserved enough in family level for making meaningful inference.

L169: the authors haven't analysed bird nesting behavior, they've analysed nest structure and location, neither of which are behaviours. They should change this to bird nests.

>> Following the reviewer's suggestion, we made changes to clarify that the study focuses on nest characters, not nesting behaviors throughout the manuscript.

L173: it would be helpful to be clearer what is meant here, without the reader having to read the papers referenced and discover what the authors of the current ms are implying here. Also see lines 175-179: what do the authors really mean here?

>> We agree that the original sentences need clarification. To also address Reviewer 2's suggestion, we rewrote the sentences and remove controversial citations (L170-175).

L183: do the known within-family data support this claim?

>> Yes, according to our nest character data obtained from HBW, 148 of all 236 (62.7%) families (excluding those whose nest character states were treated as missing data in our analyses, e.g., brood parasites) have within-family variation in nest site, but only 97 families (41.1%) have within-family variation in nest structure (Figure 1 in the manuscript). Therefore, in general most families tend to be more conserved in nest structure than in nest sites.

L192-193: could the authors say this more clearly and just as a biological statement: Interestingly, we found a negative relationship between the dissimilarity in nest attachment approaches and the phylogenetic distance across all bird families.

>> We changed the sentence to "Interestingly, our Mantel test indicated a ubiquitous fashion that distantly related families use similar nest attachment approaches." (L189-190).

L209: lack of genetic variation in what?

>> *We added "in nest attachment-related genes" in this sentence to make it clearer (L206).*

L229: what is 'complex'?

>> *To make the sentence clearer, we rewrote the sentence as "That is because compared with non-passerines, passerines tend to build nests with more elaborate nest structures (i.e., domes or cups) and attachment approaches (i.e., lateral, horizontally forked and pensile attachment). More manipulative movements are needed to complete the elaborate nests, which are more likely to require the aid of new genetic variation than relaxation from genetic constraint" (L237-241).*

L245: 'to build', rather than 'to use'; is it the structure that has evolved or the choice of material used, which might strongly affect the structure that can be built or the choice of location, which might then impact on the structure?

>> *We changed it to "build/use" because cavity nesters include those who use natural or used cavities (i.e., secondary cavity nesters). Regarding the second question, we cannot make inferences because we did not analyze nest materials (such information is often lack or inconsistent in literatures, making the character difficult to be characterized systematically) in this study. However, we agree that nesting materials is another interesting and important nest character to be investigated further in the future. Therefore, we added a sentence in the last paragraph in the Discussion (L298-299) to point out this and other potentially important characters.*

L256: what is sophisticated about a cup or a dome (see the Price and Griffith paper for domes preceding cups)? Do they require a larger number of manipulative movements in their production? A greater variety of materials?

>> *Given that we already added a similar explanation in an early paragraph (L237-241), we decided not to add repeated information here, but rewrote this sentence as "With compact nest structures (i.e., small and tightly interwoven cups and domes) that allow flexible placement of nests among vegetation, passerines intensified their use of trees as nest sites and further explored lower vegetation layers (e.g., bush and strong herbaceous vegetation)" (L273-276) to make our point clearer.*

L276: 'intentional'?

>> *To avoid confusion, we deleted this term.*

L277: does this exclude mud cups?

>> *It includes mud cups. Thus, we revised the definition by adding "or mud" in the end of the sentence (L312) to make it clearer.*

L282: why make this assumption? What is it based on (see point re L256 above)

>> *We made this assumption because domed nests with a tunnel exit require more construction steps (e.g., black-headed weavers and striated swallows build the tunnel exit of their nests at the final stage) and thus might require additional genes coding for*

the extra building behavior to complete nest building than domed nests without a tunnel. To explicitly infer the evolutionary history of nest structure, we decided to categorize them as two different types. Although Price & Griffith (2017) considered all dome nests as one single character state, they acknowledged that some dome nests have tunnel exits. In addition, other studies such as Simon & Pacheco (2005) separated dome nests with tunnel exits from many other dome nests. Therefore, we believe that our characterization approach provides a chance to explore bird's nest evolutionary history in more details.

L287: this then means that you decided on nest structure based on its function (i.e. the value or not of being inside a cavity), not on the building required. Did you actually use this decision rule for all of your classifications? Also see L321 with regard to classification of variation within the Megapodiidae.

>> We defined the nest structure types based mainly on how birds build their nests. That is also why we separated cavity nests into primary and secondary cavity nests. It is not always possible to use function to define bird nests because so far ornithologists have not fully understood the functions of bird nests. For example, although different structures imply different functions, it is still possible that similar nests have different functions or vice versa. Given that information of detailed nest structure or any nest materials inside cavities is unavailable for a good amount of birds, it is more practically reasonable to group cavity nests only to primary and secondary cavity nests based on whether the birds generate their own cavities.

Megapodiidae is obviously an exception because they build neither cavity nests nor any other structure states we defined in the manuscript. Instead, all Megapodiidae birds build similar mound or burrow (like unfinished mound) nests to bury their eggs and utilize solar or fermentation heat, rather than body heat as most birds do, to incubate eggs. Thus, we treated their nests as a special type (and we treated it as missing data in our analyses).

Overall, we use function as an auxiliary criterion for defining nest structure when available structure information is not enough to reasonably define some cases (i.e., cavity nests) or when it is difficult to define a particular nest structure type (i.e., Megapodiidae's mound nests). Although it might not be a perfect approach, we believe that it is practical and justifiable, as Reviewer 2 pointed out that our approaches (regarding the issues of the cavity nests and Megapodiidae nests) are reasonable (please see below).

L290: a very pedantic point but a nest is built and material deposited
>> Thanks for pointing this out. We changed "deposited" to "built" (L324).

L296: how did you determine the strength of the plants involved?

>> We do not need to determine the strength of the plants involved. If birds can build nests on a plant, it simply means that the plant is strong enough to support the weight of birds and their nests. We recognized that the description was somewhat redundant, so we removed it from this sentence.

L297: what is understory relative to bushes?

>> *To clarify this, we combined this with the previous sentence as “The non-tree vegetation sites included those on bushes, bamboo or thick tangled herbaceous vegetation (such as vines and reeds) that occupies forest understory, grassland or wetland habitats” (L328-330).*

L324: what data did you survey from the HBW? This suggests that you looked at species but analysed the data at the family level. Is this what you did? Its not really very clear how you dealt with multiple states within a family, especially if there was more than two.

>> *We recorded all nest character states mentioned in the family-wide descriptions in the HBW for each family. When there were two or more states of any character recorded for a family, we checked nest information at the species level to determine if more than 10% of the species in that family use a particular state. We recorded all states used by more than 10% of the species as the “major” states. For the families containing fewer than 50 species, we checked information for every species; for those containing more than 50 species, we picked up 50 species approximately evenly distributed across all genera in that family. The reason for using a threshold of 50 species was that the value is close to the mean number of specie in each bird family (10,978 species / 242 families = 45). To address Reviewer 2’s comment on the updated passerine taxonomy (please see below), we added the new families in the revised manuscript. Because family summaries for those new families were still lacking in the new version of HBW when we accessed in October, 2017, the character states were obtained at the species level for these families. We added these details in the revised manuscript (L 356-373).*

This manuscript is really missing some good illustrations of the variation the authors are seeking to explain.

Reviewer #2 (Remarks to the Author):

Until recently global analyses of avian nest evolution were handicapped by the unavailability of a complete avian phylogeny, lack of information on nests for almost a third of the world species (especially tropical), and the difficulty of accessing the existing nest data scattered through primary, often obscure, sources. All this has changed over the last decade with the unveiling of a comprehensive avian tree (Jetz et al), publication of the HBW series (where nest data are summarized for all behaviorally known species), and the avalanche of new nest data coming from the New World tropics. The authors of this manuscript, and Price & Griffith (P&G) earlier this year, are the first to reap the benefits of these new advances and data availability. Whereas P&G analyzed evolution of nest architecture across all passerines (60% of all birds of the world), the authors of this study are the first to tackle all birds of the world (at the family level). Their concept and methodology are

original and should be of interest to ornithologists and broader audience interested in animal construction and evolution of behavior.

The authors should be commended for recognizing that avian nesting behavior is multifaceted and should not be reduced to merely one or two parameters, as was done in most prior studies. The three parameters selected in this study (site, structure, and attachment) are indeed among the most fundamental and informative. The first two correspond to 'location' and 'shape' in the study of P&G. Type of nest 'attachment' is the third parameter which, in my opinion, makes a better and more independent character than 'exposure' selected by P&G. However, the authors should explicitly acknowledge that site, structure, and attachment are not the only facets of nesting behavior to consider. Future studies should include such parameters as type of material, construction method, nest ontogeny (i.e., sequence of behavioral steps taken in nest construction), etc.

>> We appreciate Dr. Zyskowski's positive comments on our study and agree that our analyses based on site, structure and attachment did not cover all facets of nesting behavior. Thus, we added such statement in the last paragraph of the Discussion (L 297-299) to address this point and bring up potential directions for further research.

In the Discussion, the authors state that "previous studies that failed to recognize the multifaceted nature of avian nesting behavior are likely to have captured only parts of the complex interactions among nest characters and thus reached inconsistent conclusions on the evolutionary trajectory of avian nests" (lines 171-174) and that treating nesting behavior as a single character "has prevented them from capturing the whole picture of avian nest evolution" (lines 211-214). Although true in the sense that no study to date, including this contribution, could capture the full complexity of avian nest evolution, these claims are overstated and unfounded. I can't think of any "inconsistent conclusions" in the four studies cited (refs 4, 18, 19, 20) and I don't see how the only example given, the recent shift in some swallows "from nesting in natural habitats to human buildings" (lines 214-219), is relevant to any of these studies. As an author of one of these studies (ref. 19, Zyskowski & Prum 1999 [not 1990, as cited]), I object to listing it as an example of "studies that failed to recognize the multifaceted nature of avian nesting behavior" (line 174). In fact, this study is based on 24 different nest characters!

>> We recognize that confusion was caused because we did not describe the conclusions of previous studies explicitly or explain our statement clearly in the original manuscript. We also feel sorry that we mistakenly cited the study. Therefore, in the revised paragraph, we rewrote the whole paragraph and explicitly discussed three potential causes of discrepant conclusions about phylogenetic conservatism of avian nests in previous studies, and what insights our study can provide to the debates (L170-172 and L207-251). The three causes are (1) investigation of different nest characters, (2) different ways of interpreting the results and (3) idiosyncratic nature of nest evolution among different lineages.

We agree that the example of recent shifts from natural nest sites to human buildings in swallows is not directly relevant to the discussion given that our analyses did not consider human buildings as a nest site. Thus, in the revised manuscript, we added a mini meta-analysis of nest site evolution in the swallow family (shown in the Supplementary Figure 1) to demonstrate our argument that focusing on different nest characters might lead to discrepant conclusions on phylogenetic conservatism in avian nest evolution.

Critical to this study are the details of character coding and scoring across avian taxa. The authors did a good job identifying broad categories of nest site, structure, and attachment. Similarly, for this rather coarse analysis, it was reasonable to: (1) treat taxa with unique forms of nesting (brood parasites, mound-builders, no nest built) as missing data; (2) code all cavity-nesters regardless of nest form as just 'cavity'; and (3) implement 10% threshold for coding polymorphisms. However, I find it problematic that avian families as defined in HBW (source of nest data) are not all the same as family-level clades in Jetz et al (source of phylogenetic trees). Here are some examples from Tables S1 & S2 of spurious coding resulting from HBW's outdated family definitions:

>> We appreciate Dr. Zyskowski's positive comments on how we categorized nest characters across avian families and how we dealt with special cases and polymorphic character states. In the revised manuscript, we followed the suggestion to update the taxonomy and nest character states for several families (please see our point-by-point responses below for details). In fact, in the version of HBW that we accessed in Oct 2017, the family-level taxonomy was already updated based on new phylogenetic evidence that matches better with Jetz et al. (2012) phylogenetic trees. Thus, we made changes accordingly in the revised manuscript. All of the taxonomic changes were in passerines, in which the number of families increased from 100 (the previous version of HBW) to 136 (the version in Oct., 2017). Major changes were found in 48 passerine families due to family level taxonomic update. That is some are new families and the others are old families with newly joined species members and/or recently removed members. We re-did all of the analyses based on the updated taxonomy and family-level nest character states. The re-analyses did not change our conclusions.

- Paradisaeidae: coded 'dome' because this family in HBW still includes Cnemophilidae

> > We agree. The latest version of HBW (Oct 2017) separates Cnemophilidae from Paradisaeidae. We updated the taxonomic status accordingly, and so the "dome" state was removed from the Paradisaeidae family.

- Sylviidae: coded 'dome' because this family in HBW still includes Cettiidae, Phylloscopidae, etc.

>> The latest version of HBW separates Acrocephalidae, Locustellidae, Phylloscopidae, Scotocercidae (the family name Scotocercidae is used to replace Cettiidae), Macrosphenidae, Bernieridae from Sylviidae, and merges previous Paradoxornithidae into Sylviidae. In the revised Sylviidae, fewer than 10% of species build domed nests and most species build cup nests. So, we coded "cup" as the effective trait (Table S1)

and included “cup” and “dome” in the all traits (Table S2) for this family in the revised manuscript.

- Turdidae: coded ‘dome’ because this family in HBW still includes Brachypteryx
>> *We agree. The genus Brachypteryx together with several genera were moved from Turdidae to Muscicapidae in the latest version of HBW (Oct 2017). Indeed, the revised Turdidae contains no species building dome nests. So we changed accordingly in the revised manuscript.*

- Eupetidae coded ‘dome’ because this family in HBW still includes Lesser Melampitta
>> *We agree. Lesser Melampitta is now placed in its own family Melampittidae in HBW (Oct 2017). The revised Eupetidae only contains one species, which uses “cup” nests. We changed accordingly in the revised manuscript.*

- Thraupidae: coded ‘dome’ because this family in HBW still includes Euphonia, Chlorophonia
>> *The two genera Euphonia and Chlorophonia are now placed in the family Fringillidae in HBW (Oct 2017). In addition, several other genera were also moved from Thraupidae to Rhodinocichlidae, Passerellidae, Phaenicophilidae, Spindalidae, Nesospingidae, Calyptophilidae, Mitrospingidae or Cardinalidae. On the other hand, several genera were moved into this revised family from Cardinalidae and Passerellidae. The revised Thraupidae contains several species (such as Asemospiza obscura, Geospiza pauper, Slender-billed Finch, etc.; more than 10% of species members) building “dome” nests, which was counted as an effective state for this family. So, we kept “dome” in both revised Tables S1 and S2.*

- Fringillidae: coded as only ‘cup’ because this family in HBW excludes genera Euphonia & Chlorophonia
>> *As Dr. Zyskowski pointed out, the revised Fringillidae also contains species (> 10%) building “dome” nest. So, we added “dome” in both revised Tables S1 and S2.*

Most families in Tables S1 and S2 are listed in phylogenetic order but there is some intermixing of passerine and non-passerine families. Since only orders are listed in the figures, it would be helpful to add a column for order into both tables and assign the families accordingly.

>> *We added a column “Order” and separated passerine from non-passerine families clearly in the revised Table S1 and S2.*

Tables S1 & S2 also include a few errors and omissions:

- Procellariidae: should not have ‘dome’ and ‘dome+tunnel’ in both tables
>> *We corrected the error in the revised tables.*

- Cathartidae: should include ‘secondary cavity’ in both tables
>> *We added “secondary cavity” to this family in the revised tables.*

- Furnariidae: should include 'dome+tunnel' in both tables

>> *We added "dome+tunnel" to this family in the revised tables.*

- Tyrannidae: should include 'pensile' attachment in both tables

>> *We added "pensile" to this family in the revised tables.*

- Emberizidae: should include 'dome' in both tables

>> *In the latest version of HBW (Oct 2017), several species were removed from Emberizidae and became new families, including Passerellidae. The revised Emberizidae now contains no species building dome nests. Therefore, we did not make the change.*

- Corvidae: should include 'secondary cavity' at least in Table S1

>> *Dr. Zyskowski is right. There are a few species in Corvidae using 'secondary cavity', and thus we added this state to the revised Table S1.*

- Pomatostomidae: attachment listed as 'pensile' in both tables, but only 1 species (Garritornis isidorei) builds pensile nests; change to 'base' in Table S1 and to 'base; pensile' in Table S2

>> *Dr. Zyskowski is correct that only Garritornis isidorei uses pensile and the other 4 species use base attachment. However, because $1/5 > 10\%$, we changed the states as "base; pensile" in the revised Tables S1 and S2.*

- Meliphagidae: attachment listed as 'pensile' in both tables, but only 2 species of Ramsayornis build pensile nests, thus 'pensile' should be excluded from Table S1

>> *Some species of this family can use both pensile and other attachment approaches. However, fewer than 10% of the species use pensile attachment in this family. So, we removed "pensile" from the revised Table S1 (effective character states).*

Although most of these details are relatively minor and probably wouldn't change the overall conclusions of the paper, it would be prudent to confirm this prior to publication. I find the conclusions of the paper plausible. The manuscript is well written and organized and the figures informative and quite handsome.

Kristof Zyskowski

>> *We really appreciate Dr. Zyskowski's careful review and his extensive understanding on avian nest characters. Making these changes significantly improved the quality of the data used in our analyses, but as expected by Dr. Zyskowski's, the revision did not change the overall conclusions.*

Reviewer #3 (Remarks to the Author):

This study looks at the evolution of several aspects of avian nests (structure,

support, site) across all avian families using a published phylogeny. Although exploration of avian nest characteristics have been conducted within several families, this has not been previously done at this level. The results allow some exploration of which characters are more labile, the order of evolution of some states, and the timing of these events allowing some comparison across the different traits. I think the overall questions are worth addressing, and some interesting conclusions were found. Others are not unexpected, however, and I think there might be more insights that could be gained (making this of greater significance to the field). Additionally, I feel there are some limitations in the methods (detailed below) that I feel limit some of the strength of the results.

First, the introduction emphasized that the availability of complete phylogenies made it possible to examine nest evolution at this scale (line 51). However, the analyses were then restricted to the family level in spite the availability of a complete (all species) phylogeny. Since many characters were polymorphic within families (and the authors had to modify the code used for one test to accommodate the polymorphism), I am not clear why the analyses were not done at the species level. The exclusion of some character states (those found in less than 10% of species) would not be an issue if all species were examined, and it would prevent rare character states from being "over-weighted". Additionally, while being over-weighted is one possibility for rare character states, there are cases where a rare character state may be biologically important to consider. For example, if an early diverging species/clade within a family exhibits a character state not found in later diverging species, it may be the ancestral condition. By excluding it in the analyses, you miss this possibility, whereas coding species would allow inclusion of this trait AND the pattern of trait changes within a family (as determined using the branching patterns of the species in the family) should allow more rigorous assessment of the ancestral states of a family. It is not clear why this more rigorous analysis was not conducted, given that it should have been possible.

>> As what we have answered to a similar question from Reviewer #1, we conducted family-level analyses in this study mainly for three reasons: (1) the completeness of nest character data, (2) phylogenetic uncertainty and (3) better trait dissimilarity measures due to within-family variation.

First, the information of nest characters is not available for every species (a fair amount of species have no nest information in literatures). By contrast, the HBW provides explicit descriptions of nests at the family level, which allow us to characterize and compare nest characters across all bird families. Nest characters at least for nest attachment and structure are conserved enough at the family level to making robust inferences, as Price & Griffith (2017) have suggested. Thus, while analyses at the species level will unavoidably suffer from missing data, family-level analyses can provide more robust results.

Second, the phylogenetic and taxonomic uncertainties can dramatically increase below the family level (such as the somewhat uncertain relationship within passerines pointed out by the reviewer below). The uncertainties may cause biases especially in

testing phylogenetic signal. Although the avian family phylogeny is also subject to change, its uncertainty is much lesser than species or genus phylogeny.

Third, with variation within the analytic unit ("family" in our study), we can better measure trait dissimilarity among the units in the Mantel tests based on categorical trait data. The within-unit variation allowed the dissimilarity to be measured at a nearly continuous scale.

To take into account the potential biases caused by the 10% threshold, we conducted the analyses on two datasets, one contains only effective character states (i.e., those found in > 10% species) and the other contains all character states mentioned in the HBW. Both datasets led to similar results, suggesting that rare character states should not be a concern in our study. Furthermore, even using species-level data, we still expect some polymorphism problems because species can also exhibit variation in nest characters, especially for nest sites. In fact, we found that well-studied species are more likely to be recorded to be polymorphic compared with understudied species. That is variations in bird nests tend to be underestimated at the species-level.

Considering the trade-off between focusing on a coarser phylogenetic level (i.e., family) with complete data and a finer phylogenetic level (i.e., species) with many data gaps and more uncertain phylogenetic relationships (and other factors discussed above), we believe that family-level analyses can provide more robust and meaningful conclusions.

Second, although phylogenetic uncertainty was accounted for in some ways, by downloading 1000 trees, these were then used to make a single consensus tree to use for character state reconstruction. The more rigorous approach would be to analyze each of the 1000 trees (BayesTraits should allow this to be done), and then use those results to assess reconstructions and look for patterns. Obviously this is more challenging (summarizing the 1000 sets of results), but it also can provide much greater information about the confidence of character state reconstruction at a node. For example, in a consensus tree, some clades may be the "best" that can be found, but still may have shown up in only a very small percentage of trees (I have seen consensus trees where a clade was only found in 2 or 3 % of all trees). When a consensus is used, this node is considered to be equal in confidence to a node that was found in 100% of all trees, even though that poorly supported node is not even found in a majority of the trees. This is most likely to affect reconstruction of traits during the rapid radiations (nodes in the gray shaded regions of figure 2), and that is where several of the focal nodes (labeled with letters) are located, suggesting some conclusions may be based on reconstructions that may have low confidence.

>> We feel sorry that some confusions were caused by our unclear method description for the ancestral state reconstruction in the original manuscript. In fact, the BayesTraits analyses we conducted were based on the 1000 individual trees downloaded from birdtree.org, not on the single consensus tree. The MCMC algorithm of BayesTraits sampled throughout the parameter space containing the 1000 trees. To

clarify this, we reorganized the paragraph describing the analysis in the Methods and added more descriptions to explain that the consensus tree was used to summarize and visualize the BayesTraits results, but not used to run the BayesTraits analyses (L 376-398). We also re-wrote the relevant sentences in the Introduction (L 73) and Results sections (L 109-110).

We understand that a consensus tree cannot present all possible nodes included in the original 1000 trees. However, a consensus tree presents the most possible tree topology (and the most efficient visualization approach) to summarize the 1000 trees' ancestral reconstruction results. In fact, no any single tree or figure can show all possible nodes given that there are so many possible tree topologies for a tree with 242 terminal tips. It is also not practically reasonable to show all 1000 trees with ancestral trait states in the manuscript, and this is not how BayesTraits is designed to deal with phylogenetic uncertainty. Since we used the majority rule to generate the consensus tree, each branching pattern can be found on at least 50% of all trees, and thus this consensus tree shows the most possible clades. Although this approach is not perfect, it is the best one we know (and most studies use) to present the results of ancestral state reconstruction.

Specific comments:

Line 30: I was unclear what phylogenetic coverage meant, but I presume this means level (among orders versus within a single order?).

>> Yes, it meant phylogenetic level. To address Reviewer 1's comment, we rewrote this sentence and removed the term from the sentence (L29-31).

Line 51: Two of the three citations given to represent "complete phylogenies" are not very complete, and instead include only 0.5% or 2% of all avian taxa. Only Jetz et al. is complete. There are other complete, or more complete phylogenies that are published for birds that would be more appropriate to cite here (Holt et al. 2013, Burleigh et al. 2015) if a desire is to show that trees encompassing a large amount of avian diversity are available.

>> We appreciate the reviewers' suggestion and agree with it. Thus, we replaced two original citations with Holt et al. 2013 and Burleigh et al. 2015. In addition, we said "nearly complete" in the revised sentence because no any published avian phylogeny so far contains every single species due to sampling gaps and continuously updated taxonomic statuses (L48).

Line 69: I might rephrase this as "across all modern bird families", given that this is not looking at patterns within families (unlike the studies cited in line 68), but instead at a coarser level.

>> We agree with the reviewer and thus changed the sentence following the suggestion (L67).

Line 116-117: What is the confidence about the assertion that secondary cavity nesters evolved first? This occurs during the rapid radiation, and it appears that

primary cavity nesting shows up very shortly after secondary. This is an example where a more rigorous treatment of the 1000 trees might yield some different conclusions (or reduce confidence in the conclusion).

>> As we clarified earlier, the ancestral state reconstruction was based on the 1000 phylogenetic trees, and thus the phylogenetic uncertainty was accounted for in this analysis. We agree with the reviewer that in general rapid radiation may increase the uncertainty in phylogenetic inference. However, our statement on the evolutionary order of the two character states was based on the mean probabilities estimated across all 1000 phylogenetic hypotheses. Therefore, we think that the result is rigorous. In addition, this specific order is not the main focus of this study and thus we just simply mentioned the result from our analysis, but not further interpreted it in our manuscript.

Line 117-118: Is it possible to actually estimate the number of gains of cup nests (using a pre-defined criterion)? The arrow D points to two nodes, suggesting that 3 is most likely (one outside of passerines, and two noted here). If the evidence is more equivocal, perhaps give a range (3-10 times). That would make it easier to evaluate how labile this trait may be for readers.

>> Although it is possible to determine the number of gains of cup nests in the evolution of avian families, the evidence is equivocal. Following the reviewer's suggestion, we provided a possible range (3 - 16) for the independent appearances of cup nests in the revised manuscript (L 119-121). There are at least 3 (one in non-passerines and two in passerines) initial gains of cup nests, and there can be another 13 more recent gains (3 in non-passerines and 10 in passerines).

Lines 132-139: This raises some other interesting questions. What about loss/reversibility (e.g., are there some character states do not appear to be reversible, when a character state changes is it more likely to revert to an earlier state or go to a novel state)? The focus on what is presented is what might precede other things, but there are other insights that might come from this analysis that I encourage authors to explore.

>> To address the comment, we added a paragraph for discussing the possible reverse evolution of cup nests in the Discussion section (L 282-289). The argument is also relevant to another study, Price & Griffith (2017), published early this year. Our results showed that cup nests evolved before passerines occurred. Thus, the cup nests in passerines may be a result of (1) reverting to an earlier state that already showed in the lineage of hummingbirds, swifts and treeswifts or (2) changing to a similar state with non-homologous genetic basis. On the other hand, we avoided making too many inferences out of the results because such inferences may not have high confidence (as the reviewer raised similar concerns in the letter).

Line 148: In the preceding paragraph, there is some directionality that is pointed out (what states precede other states). While not all traits may have directionality, or within a trait not all states may work in a directional way, it is clear there is some directionality (and it is easy to see how some states are likely to have evolved from other states, leading to directionality in some cases). Therefore, doing an analysis

assuming non-directionality may not be a good fit to the data. Can the appropriateness of this assumption be explored or tested to ensure that these conclusions are not biased by assumption violations?

>> We think that we probably did not make this clear in our original manuscript. We did not fit any non-directionality model to our character state data. What we did was to use a neutral model to simulate trait states for each tip (i.e., family) as possible character state distributions under a null hypothesis. We then contrasted the empirical data on character states to the simulated data to test whether the empirical data was significantly deviated from the null simulations (i.e., whether the empirical data showed directional trait evolution or, more precisely, strong signal of phylogenetic conservatism). To clarify this, we rewrote the sentence to clearly state that the neutral model was used as a null model (L 146-147).

Line 173-174: Given that the cited studies were restricted to within families, where the range of states may be lower (and the power to assess patterns may be lower), could that also explain the inconsistencies that are referred to?

>> To address one of Reviewer 2's concern, we removed the citations from this sentence and rewrote it to a more general description in the revised manuscript. Yet we agree with the reviewer that difference in phylogenetic levels of focal data is another reasonable explanation for the inconsistent patterns in nest evolution. Therefore, we added this alternative in the next sentence (L 172-175).

Lines 182-183: I am not clear why nest structure would exhibit both greater directional selection and greater conservation, at least if I understand the meaning of what is being said here. It would seem that these are opposite (if there is high directional selection, then it would suggest low conservation).

>> We recognize that such argument may cause confusion. However, strong phylogenetic signal (or phylogenetic inertia) can mean conserved evolution between closely related taxa or continuous evolution of traits towards a particular direction or both (Blomberg & Garland 2002). We concede that the most straightforward interpretation of a strong phylogenetic signal is phylogenetic conservatism. To avoid the confusion, we removed the statement of "directionality" from this sentence given that the change will not obscure the main point of this paragraph and rewrote the sentence (L179-180).

Line 224-226: Could errors in the passerine tree be an alternative explanation as well? Oscine passerine families are still not completely well understood, but it is known that traditional classification of families (which underlies parts of the Jetz et al. phylogeny) are often quite different from families circumscribed using molecular data.

>> We think this suggestion is insightful and thus added such argument in this paragraph (L 241-246). Although incorrect taxonomic status of passerines may cause biased estimation of phylogenetic signal, we believe that it is not the main explanation for the pattern observed in this study. That is because our inference was based on the comparison of two datasets (i.e., all avian families and only passerine families), which include the same passerine data, and thus such biased effect should be somewhat

canceled out when we compare their phylogenetic signals.

Lines 304-315: How were cavity nests coded for attachment site?

>> The attachment of cavity nests was coded as basal attachment because such nests are mainly supported from their bottom. This was described in Methods (L 338-340).

Lines 325-328: I applaud coding polymorphisms by providing the different character states, and not selecting the most common state to represent a family. However, if the number of species within a family exhibiting each state is not known, how can it be known whether 10% of species exhibited a particular character state or not? Much greater clarification of what this means and how it was calculated should be provided.

>> We think the reviewer pointed this out. The description for the coding was not detailed enough in our original manuscript. We recorded "all" nest character states mentioned in the family-wide descriptions in the HBW for each family. When there were two or more states of any character recorded for a family, we checked nest information at the species level to determine if more than 10% of the species in that family use a particular state. We recorded character states used by more than 10% of the species as the "effective" states. For the families containing fewer than 50 species, we checked information for every species; for those containing more than 50 species, we picked up 50 species approximately evenly distributed across all genera in that family. The reason for using a threshold of 50 species was that the value is close to the mean number of species in each bird family (10,978 species / 242 families = 45). To address Reviewer 2's comment on the updated passerine taxonomy (please see our responses to Reviewer 2), we added new families and updated nest data for old families with changes in their species members in the revised manuscript. Because family summaries for the new families were still lacking and those for the revised families were out of date in the new version of HBW when we accessed in October, 2017, the character states were obtained at the species level for these families. We added these details in the revised manuscript (L 356-373).

Line 339: I think it may be clearer to just indicate you used the trees based on the Hackett constraint (for those not familiar with the bird tree, just indicating Hackett constraint in parentheses may be confusing).

>> We changed it following the reviewer's suggestion (L 377).

Line 339: The Hackett et al. tree differs from the Jarvis and Prum trees in several ways, not just Coliidae (please note spelling) - and even the Jarvis and Prum trees differ from each other. Therefore, I do not see why Coliidae was excluded, yet other likely conflicting placements were retained. Possibly the other conflicting placements were not included in the constraints used by Jetz et al. (some well-supported relationships in Hackett et al. were not included in the constraint tree), but if so please state you carefully checked all Jetz et al. relationships against Jarvis and Prum, and excluded all families that were in conflict to ensure the same rules for inclusion in the tree were applied equally across the tree.

>> The reviewer is right that the three published phylogenies (i.e., Hackett et al. 2008, Jarvis et al. 2014 and Prum et al. 2015) are inconsistent with one another for more than one family. We were not trying to exclude all inconsistent families from analyses because (1) there would be too many families being excluded and (2) every phylogeny is just a hypothesis (given that a true history is generally unknown) and thus a reasonable level of inconsistency is expected. We removed Coliidae in the original manuscript because its positions in most of the 1000 sampled trees are in conflict with all of the three above studies. Other families are at least consistent with one of the three studies. However, to address the reviewer's concern, we put the Coliidae back to the analyses in the revised manuscript, and the results were generally consistent with the previous ones. The only change is that adding Coliidae increases one case of recent, independent gain of cup nests in non-passerines, which is already equivocal (see our discussion for the reviewer's comment on Line 117-118) and thus barely affects our conclusions.

Line 343-344: Was the consensus tree made in phytools? (given how it is written, it is not clear). More importantly, was the method used to generate the consensus one that maintained branch length information (and made consensus branch lengths) or not (most consensus methods only retain relationships, and ignore branch lengths). This should be mentioned explicitly as branch length treatments can have profound effects on the results.

>> Yes, we used phytools to generate the consensus tree (Majority-rule consensus tree) with branch length estimated using the "consensus.edge" function with a "least square" approach. We added the detailed information in Methods section (L 395-398).

Lines 347-349: Was convergence assessed in some way? This is important to demonstrate.

>> We ensured the convergence by reaching the acceptance rate of 20% - 40% (a value suggested in the manual of BayesTraits) and repeated the analyses 3 times to check the consistency in parameter estimates among different runs. We added this part in Methods (L 383-388) and put figures showing convergent trends of example estimates from 3 independent runs in the supplementary information (Supplementary figure 2).

Another point to consider might be to try and look more rigorously at the potential interdependence of states. As stated in the discussion (e.g., line 179), the traits are interdependent to at least some degree. For example, it is hard to see how a non-basal attachment could be functional for a scrape nest. However, these analyses focused on a single trait at a time without any assessment of any interdependence (except in comparing results of analyses of one trait with results of analyses on another trait). It might provide a more robust picture of how nests evolve over time to try and look at traits together to explore the interdependence (though I admit, off the top of my head I am not certain the best approach for doing this, other than doing an evolutionary PCA in phytools).

>> We have considered this issue seriously. However, as the reviewer pointed out that there is no proper approach to quantify the interdependence among different characters in one single, integrative analysis. Furthermore, most approaches used to assess interdependence or correlations among traits require raw data with continuous (such as evolutionary PCA) or binomial distributions, not multiple categories (that is our data type). Many binomial-based approaches can only exam one state for each character in one time. Therefore, the analytic results of such approaches are hard to interpret because they are still separated, rather than integrative, information. Although our approach cannot directly provide quantitative estimate of the interdependence of characters, it does offer comprehensible conclusion based on statistically supported phylogenetic history. That is our results show straightforward, temporary-ordered patterns for the evolution of all three characters, facilitating our interpretation in their potential interdependence.

The discussion suggests the role for nest structures as leading to adaptive radiations in birds. While this may be true, there are not explicit analyses to address this, nor are any verbal arguments of alternative hypotheses considered. Thus, while I think this is worth retaining, it should be clear it is one possible explanation and I might reduce a little emphasis on this point.

>> We agree with the reviewer's comment, and thus added a discussion that the inferred relationship between the evolution of birds and their nests in the last paragraph is only one possible hypothesis, and further studies are warranted (L 292-294). We also discussed potential future research that could help to answer the question (L294-297).

Reviewers' comments:

Reviewer #1 (Remarks to the Author):

The authors have addressed my comments and I have only a handful of other comments to make on the revision. The biggest of which is the assumption that all nest building is genetic when, in fact, there are no data to support this supposition. I think the authors should be more circumspect with their enthusiasm throughout about the basis of all these nest attributes. There is for example never any comment about human tools being genetic even though they stayed the same over a very long period of time. Indeed, stereotypy of tools (those made by crows and all primates) is considered evidence of planning.

My second major point is that the writing could be made much tighter in many places. For example, the first line of the abstract says nothing at all and should be deleted, indeed, it is not entirely clear what a naïve reader would take away from the abstract. Perhaps the Editorial staff can offer some help here.

Specific comments in line order:

L 61-68 actually the data haven't been sufficiently closely examined to claim this

L 69: The work presented here is not a novel view but rather an analysis bigger than any yet conducted.

L86: Does this mean that most families build cup nests?: cup nests were most prevalent across modern bird families, followed by secondary cavities, domed and platform nests.

L89: here and elsewhere: build/make nests rather than use nests?

L95: no evidence for 'can', so delete

L119-121: this is a bit circular

L133: is this not necessarily the case: For example, only after platform or cavity nests evolved from scrape nests (B in Fig. 2a), did the ancestral nonpasserines start nesting on other nest sites, including trees, water bodies and cliffs/banks (A, B and C in Fig. 2b, respectively). Ditto the example on L138

L149: Describe in biological rather than statistical language

L180-2: I don't think this is suggested by these data at all: the nest structure tells us very little about construction skills, nothing about whether they are complex and nothing about genetic constraints.

L191: what is constrained here? The only/best way to make an attachment? Or a genetic constraint – this latter seems nonsensical.

L204: do you really think 'can only' – isn't it the case that it is structurally more effective to do things this way – attaching a structure in the most useful way does not mean it is the only way to do it.

L239: Although this is a common assumption, I don't know of any quantitative data to support this claim so it would be useful to include a reference.

Reviewer #3 (Remarks to the Author):

This version of the manuscript (and the response from the authors to previous comments) addresses many of the questions I had raised initially, and gives me greater confidence in the results. There are still a few clarifications that I think should be made and I have a few other suggestions.

Throughout the text, the authors use scientific names of large, multi-ordinal groups (Coraciiformes, land-birds) and common names. In contrast, the figures are labeled with orders, and so lack both common name and larger groupings. That is fine if the only target audience is one that is very knowledgeable about avian systematics, but might be limiting for some other readers that may struggle with linking what is pointed out in the text with what is going on in the figures since different terminologies are being used. I encourage the authors to think about whether there might be a better solution that would make the manuscript more accessible to a wider audience.

Line 50: Rather than comprehensive (since this study does not sample comprehensively at the species level) I might suggest using the term "broad" here.

Line 56-58: This idea of a putative "syndrome" is not explicitly referred to again in the discussion (or results, though the relevant information is discussed). I might either delete the use of the term here, or (which I prefer) explicitly utilize the term syndromes in the discussion (in essence, that nest characteristics may not form a syndrome).

Line 60-61: I was unclear on the meaning of the phrase "avian species build their own specific nests, suggesting a significant genetic basis for their nesting behavior" (I am not clear on what the first half of it means, and also not sure how that might relate to the last part of the quote).

Line 95: At the end, should that be nest site types rather than just nest sites?

Line 120-121: "although the evidence could be more equivocal" suggests (to me at least) that the evidence is even more vague than the range 3-16 suggests. Is that the meaning the authors want to convey (and if so, would it be clearer to write "although the evidence could be even more equivocal"). If that is not the intended meaning, then I would suggest rewriting this.

Line 133: Interdependence among all three characters is challenging, though some limited testing could be done using something like a concentrated changes test (or Pagel's 1994 test) if the focus was on two traits and two specific character states. I could see why the authors might not want to do some of those tests, but it would be a way to more rigorously examine some of these key results.

Line 189: Should this be "indicated IN a ubiquitous fashion"? I would argue ubiquitous in this context is misleading - although most species use one attachment type (so it is ubiquitous among families), in terms of evolutionary events, this has not arisen many times but instead is a shared, ancestral trait. The sentence as currently written suggested to me that there may have been many repeated gains of some attachment types which I do not feel is supported by the figure (or what is written in the remainder of this paragraph).

Line 258 and the remainder of the paragraph: This is speculative (and is written clearly to indicate that). It would be possible to more rigorously test some of these ideas (e.g., BISSE if you reduce to binary, or MuSSE if characters are kept as multi state) and this would strengthen the story.

Line 361: As written, the implication is that nest type information existed for every species, yet it is clear in your response to previous comments that this is not true (and is the main reason analyses were not done at the species level). I would suggest including here what was done when there was a lack of nest information for some species in a family.

Line 377: Arbitrarily selecting one species (if it truly was arbitrary) could lead to cases in which Jetz is incorrect. Since about 1/3 of the species included in Jetz lacked underlying sequence data, their position in the trees is based upon traditional taxonomy. In some cases, more recent data has shown that there are taxa that are misplaced into the wrong family through this procedure (since taxonomies can be wrong). Was there any attempt to check the "arbitrary" selection to ensure that there was not likely an error in these?

Responses to Reviewers' Comments

We thank the reviewers for their time and effort in helping us improve our manuscript. Our answers for each comment/question are shown after the symbol ">>" and in *italic*.

Reviewers' comments:

Reviewer #1 (Remarks to the Author):

The authors have addressed my comments and I have only a handful of other comments to make on the revision. The biggest of which is the assumption that all nest building is genetic when, in fact, there are no data to support this supposition. I think the authors should be more circumspect with their enthusiasm throughout about the basis of all these nest attributes. There is for example never any comment about human tools being genetic even though they stayed the same over a very long period of time. Indeed, stereotypy of tools (those made by crows and all primates) is considered evidence of planning.

>> *We understand the reviewer's concern that we do not have direct evidence that genes regulate avian nesting behavior based on experiments, such as genetic functional tests. Thus, we reduced genetic-relevant statements at several points in Abstract (Line 21), Introduction (Line 61) and Discussion (Line 183-185) in the revised manuscript to avoid the impression that nesting behavior is completely determined by genes. However, this does not mean that the assumption about the genetic basis of nesting behavior is wrong. In fact, evidence for the genetic basis of behavior does not necessarily need direct genetic functional tests, and most of such tests are not technically possible now. (Darwin inferred that organisms could pass down traits to their offspring without knowing the existence of genes.) We do not mean to initiate arguments relatively irrelevant to this article. However, regarding the example of tool-using behavior, we have different points-of-view from that of the reviewer. Tool-using behavior is only observed in a few species of mammals and birds—such as humans and crows—and most mammalian and avian species cannot learn to use tools. In fact, many species have been tested in lab for their ability to use tools and they fail in the tests. This is evidence that tool-using behavior has its genetic components that determine a species' capacity to develop such a behavior. Following this rationale, we think that nesting behavior should also have its genetic basis as some avian species are able to build very complex nests and some are not. We also believe that the phylogeny-based methods we used are among the possible approaches to examine the question, at least to some level. It is important to note that this argument does not deny a role for experience or learning in nesting behavior. We think that genes might impact some nest characters that show strong phylogenetic signal, such as nest structure, more than others with more labile evolutionary patterns, such as nest site. This is what we try to argue throughout this manuscript. Whether nesting behavior can be genetic-based is a long-term debate, and we think the arguments we make in the manuscript can provide some insights into it.*

My second major point is that the writing could be made much tighter in many places. For example, the first line of the abstract says nothing at all and should be deleted,

indeed, it is not entirely clear what a naïve reader would take away from the abstract. Perhaps the Editorial staff can offer some help here.

>> *We appreciate the comment. In order to make the writing concise and clear to a naïve reader, we had the manuscript been edited by a native English speaker who is not in the field of evolutionary biology.*

Specific comments in line order:

L 61-68 actually the data haven't been sufficiently closely examined to claim this
>> *We make several arguments in this paragraph and it is not clear to us which claim is not closely examined according to the reviewer's comment. The main question we want to test here is whether modern bird families show strong phylogenetic signals in their nest characters. The phylogeny-based analyses we conducted in this study are designed to answer this question, and so we believe that we have closely examined it. According to the reviewer's comments in the early part of this letter, we think that the reviewer's concern is whether nesting behavior is genetic-based or not (the first sentence of this paragraph). Thus, we rewrote the first sentence to address this concern and another reviewer's (Reviewer #3) comment that it is unclear. We changed this sentence to "Birds of the same taxonomic groups are often observed to build similar nests. For example, almost all pigeon species build flimsy, shallow nests." (Lines 61)*

L 69: The work presented here is not a novel view but rather an analysis bigger than any yet conducted.

>> *To make the writing more clear, we rewrote the first two sentences of this paragraph and we did not use the term "novel" in the revise manuscript.*

L86: Does this mean that most families build cup nests?: cup nests were most prevalent across modern bird families, followed by secondary cavities, domed and platform nests.

>> *Yes. To avoid the confusion, we changed "prevalent" to "common".*

L89: here and elsewhere: build/make nests rather than use nests?

>> *Done*

L95: no evidence for 'can', so delete

>> *We deleted it.*

L119-121: this is a bit circular

>> *Considering that the inference is not critical to this article and it may cause confusion (as it did to the two reviewers), we decided to delete this sentence.*

L133: is this not necessarily the case: For example, only after platform or cavity nests evolved from scrape nests (B in Fig. 2a), did the ancestral nonpasserines start nesting on other nest sites, including trees, water bodies and cliffs/banks (A, B and C in Fig. 2b, respectively). Ditto the example on L138

>> *Some avian nests in nature show that this is not necessarily the case. For example, potoos and falcons use trees and cliffs/banks, respectively, as nest sites but make scrape nests. In addition, Reviewer #3 apparently agrees that this is not necessarily the case because s/he asked us to conduct additional statistic tests for these inferences. Thus, in*

the revised manuscript, we added analyses to test the interdependence among the nest characters using the BayesTraits program, and the results confirmed the interactions between the nest characters (see details in our response to Reviewer #3's comment and the revised manuscript (Lines 132-152)).

L149: Describe in biological rather than statistical language

>> To address the comment, we added some biological interpretation after this sentence. We added: "That is there was a stronger resemblance between nest structures from birds of closely related families than the evolutionary distances suggested." Nevertheless, we think that it is necessary to show the results of statistic tests in the Results section, and most (biological) interpretations should be put in the "Discussion" section (as we did), although we agree that adding a bit of interpretation right after results may help readers to understand the meanings of the results. In fact, we already briefly mentioned the general meanings of these statistical tests in the second sentence of this paragraph. We believe that with the added sentence, readers can better understand the biological meanings of the results.

L180-2: I don't think this is suggested by these data at all: the nest structure tells us very little about construction skills, nothing about whether they are complex and nothing about genetic constraints.

>> We disagree with this comment. If the structure of nests reveals little information on nest construction skill, we do not think other characters can reveal more. Studies, such as Hall et al. 2013 in Biology Letters (we added this citation, Reference 28, in this paragraph), have also pointed out that nest structure may reflect the manipulative nest-building behaviors of birds. When you compare the complex domed nests of weaver birds with the simple platform nests of pigeons, it is clear that weavers have more complex nest construction skill than pigeons. The only better data than nest structure we can think about is the direct observation of birds' nest construction behavior, which is not available for most avian species in nature.

In addition, when the EM-Mantel test results show stronger phylogenetic conservatism in nest structure, it means that closely related birds build nests that are significantly more similar in structure than expected based on evolutionary distance. Genetic (or evolutionary) constraint is the most reasonable explanation for such pattern. Thus, we believe that our arguments here are justifiable. Actually the last comment (for L239) of this reviewer suggests that he or she recognizes that genetic regulation is a common assumption for affecting nest structure and nest building behavior.

L191: what is constrained here? The only/best way to make an attachment? Or a genetic constraint – this latter seems nonsensical.

>> This paragraph discusses the possible reasons why nest attachment approaches other than basal attachment occurred at low frequencies in birds. To avoid confusion, we changed "constrained" to "conserved". As we discussed in the paragraph, the pattern could result from 3 possible conditions, "strong benefits of a common state, functional interdependence with other characters, and/or insufficient genetic variation" (Lines 215-217). In fact, the "only or best" way (mentioned by the reviewer) may not be a proper explanation for the constraint evolution. Apparently the most common character state, basal attachment, is not the "only" character state found in nature; "best" is an unclear

term, and “more effective” should be used instead (we discuss this in the reviewer’s next comment). The conserved pattern also may be caused by limited genetic variation. As we have discussed throughout the rebuttal letter and the manuscript, we believe that genetic regulation is one possible, not nonsensical, mechanism to influence nesting behavior evolution.

L204: do you really think ‘can only’ – isn’t it the case that it is structurally more effective to do things this way – attaching a structure in the most useful way does not mean it is the only way to do it.

>> *We totally agree with the comment and thus changed the sentence to “Non-basal attachment approaches are mainly restricted in passerines probably because they are more effective when being used with domed or cup nests than other structures”.*

L239: Although this is a common assumption, I don’t know of any quantitative data to support this claim so it would be useful to include a reference.

>> *As we know, there is no study so far directly proving that the more manipulative movements used for building more elaborated nests require new genetic variation in birds. However, there is indirect evidence proper for formulating the hypothesis. For example, Hall et al. (2013) show that birds building more complex nests have higher level of foliation in their cerebella and Chen et al. (2015) show that novel genetic variation contributes to the complex surface of human brain. Thus, we added the two references to the sentence and rewrote it as “More manipulative movements are needed to complete the elaborate nests, which are found to be associated with higher levels of cerebellar foliation (Hall et al, 2013) and thus are more likely to require the aid of new genetic variation than relaxation from genetic constraint (Chen et al. 2015)”.*

Reviewer #3 (Remarks to the Author):

This version of the manuscript (and the response from the authors to previous comments) addresses many of the questions I had raised initially, and gives me greater confidence in the results. There are still a few clarifications that I think should be made and I have a few other suggestions.

Throughout the text, the authors use scientific names of large, multi-ordinal groups (Coraciimorphae, land-birds) and common names. In contrast, the figures are labeled with orders, and so lack both common name and larger groupings. That is fine if the only target audience is one that is very knowledgeable about avian systematics, but might be limiting for some other readers that may struggle with linking what is pointed out in the text with what is going on in the figures since different terminologies are being used. I encourage the authors to think about whether there might be a better solution that would make the manuscript more accessible to a wider audience.

>> *We appreciated the useful suggestion. To make the text and the labels in the figures more consistent, we add the order names after the common names of species in the text (in Line 92, 118, 125-127, 131, 199-201, 210, and 305) that refer to the figures so that readers can connect the information from the two parts more easily. In addition, we added a supplementary figure (Fig. S5) in the revised manuscript to label the names of all 242 families on a phylogeny tree, which is the same tree shown in Fig. 2. So that*

readers can check the nest information in Fig. 2 more easily.

Line 50: Rather than comprehensive (since this study does not sample comprehensively at the species level) I might suggest using the term "broad" here.

>> *We change the term to "broad".*

Line 56-58: This idea of a putative "syndrome" is not explicitly referred to again in the discussion (or results, though the relevant information is discussed). I might either delete the use of the term here, or (which I prefer) explicitly utilize the term syndromes in the discussion (in essence, that nest characteristics may not form a syndrome).

>> *We agree with the reviewer's comment and added one sentence in the first paragraph of Discussion (Line 185-187) to address this point. The added sentence is 'Bird nest characters should not be treated as a character "syndrome" because they do not change synchronically across the avian phylogeny.'*

Line 60-61: I was unclear on the meaning of the phrase "avian species build their own specific nests, suggesting a significant genetic basis for their nesting behavior" (I am not clear on what the first half of it means, and also not sure how that might relate to the last part of the quote).

>> *To avoid the confusion, we deleted this sentence and added two sentences to address our points. The two sentences are " Birds of the same taxonomic groups are often observed to build similar nests. For example, almost all pigeon species build flimsy, shallow nests."*

Line 95: At the end, should that be nest site types rather than just nest sites?

>> *We agree and changed it to "nest site types".*

Line 120-121: "although the evidence could be more equivocal" suggests (to me at least) that the evidence is even more vague than the range 3-16 suggests. Is that the meaning the authors want to convey (and if so, would it be clearer to write "although the evidence could be even more equivocal"). If that is not the intended meaning, then I would suggest rewriting this.

>> *Considering that the inference is not critical to this article and it may cause confusion to readers (as it already did to the two reviewers), we decided to delete this sentence.*

Line 133: Interdependence among all three characters is challenging, though some limited testing could be done using something like a concentrated changes test (or Pagel's 1994 test) if the focus was on two traits and two specific character states. I could see why the authors might not want to do some of those tests, but it would be a way to more rigorously examine some of these key results.

>> *We followed the suggestion to conduct the analyses and the results supported our arguments. That is we examined the interdependence between pairs of the three characters using the program, BayesTraits (Pagel 1994, Pagel & Meade 2004). Given that the function in BayesTraits can only test binary states, we converted the focal characters from multiple states to binary states for the analyses. Specifically, we tested the three example arguments on character interdependence mentioned in the paragraph. First, we tested the interdependence between scrape structure (scrape vs. non-scrape) and ground site (ground vs. non-ground) and examined whether birds started nesting on*

sites other than ground after they changed nest structure from scrape to other types. Second, we tested the interdependence between cup structure (cup vs. other structure types) and non-tree-vegetation site (non-tree vegetation vs. other site types) and examined whether non-tree vegetation was used as a nest site after birds built cup nests. Finally, we tested the interdependence between basal attachment (basal vs. other attachment types) and domed/cup nests (dome/cup vs. other structure types) and examined whether non-basal attachment approaches evolved after the lineages started using domed or cup nests.

In general, the dependent models were strongly supported compared with the independent models in all of the three cases. In addition, the transition directions among character states estimated from the dependent models were consistent with our inferences based on the ancestral state reconstruction. We reported the results in the Results section (Line 132-152) and a supplementary figure (Fig. S1), described the detailed approach in the Methods (Line 425-453), and added one relevant point in the Discussion (Line 222-223).

Line 189: Should this be "indicated IN a ubiquitous fashion"? I would argue ubiquitous in this context is misleading - although most species use one attachment type (so it is ubiquitous among families), in terms of evolutionary events, this has not arisen many times but instead is a shared, ancestral trait. The sentence as currently written suggested to me that there may have been many repeated gains of some attachment types which I do not feel is supported by the figure (or what is written in the remainder of this paragraph).

>> *We appreciate that the reviewer pointed out this unclear point. We deleted "in a ubiquitous fashion" and rewrote the sentence as "the Mantel test results indicated that avian families' nest attachment approaches might be more similar to those of distantly related families than more closely related one."*

Line 258 and the remainder of the paragraph: This is speculative (and is written clearly to indicate that). It would be possible to more rigorously test some of these ideas (e.g., BISSE if you reduce to binary, or MuSSE if characters are kept as multi state) and this would strengthen the story.

>> *We understand that the discussion is somewhat speculative, and thus we stressed that it is still a hypothesis in the last paragraph of the Discussion. We also appreciate the help from the reviewer with a possible approach to strengthen our argument. However, the suggested approaches do not well fit our need. The MuSSE (or BiSSE) is used to test the effect of character states on the speciation rates averaged across the evolution history (the whole phylogeny), but our argument is that new nest structure types contributed to the adaptive radiation that occurred at a particular time point or a relatively short period of time (e.g., around 65 million years ago for the first adaptive radiation in modern bird evolution). However, despite the concern about the suggested approach, we still conducted MuSSE tests (we provided a separate document, named "MuSSE methods and results", with detailed methods and results of the tests). As we expected, the results cannot help to demonstrate our point. In brief, we grouped the nest structure states to three categories: scrape, platform/cavity, and cup/dome, because we argued that the first adaptive radiation was associated with the evolution of non-scrape nest structure and the second one was associated with the evolution of cup/domed nests. The average speciation rate in avian lineages with scrape nests is the highest, followed by that of*

cup/dome nest and then that of platform/cavity nests. Since the estimated speciation rates are for the whole phylogeny, the results tell us little about whether the platform/cavity and cup/dome nests led to bursts of new species (or families) at the period of time when these structure types emerged for the first time. Due to the inappropriateness of this approach, we decided not to include the MuSSE analyses in the manuscript. However, we can add them as Supplementary Information if the editor sees the need.

Line 361: As written, the implication is that nest type information existed for every species, yet it is clear in your response to previous comments that this is not true (and is the main reason analyses were not done at the species level). I would suggest including here what was done when there was a lack of nest information for some species in a family.

>> We are sorry for not making the point clear enough. When the nest information is lacking for a species, we exclude that species and find data from another species in that family if possible. So the threshold of 50 species is based on species with available nest information in the HBW, not the total number of species in the focal family. Thus, the correct statement for the sampling approach should be “For the families in which fewer than 50 species had nest information available in the HBW, we used the information from as many species as possible. For those containing more than 50 species with nest information available, we used the information from 50 species approximately evenly distributed across all genera in that family and estimated the percentage of each nest character state.” We rewrote the method as above (Line 381-386).

Line 377: Arbitrarily selecting one species (if it truly was arbitrary) could lead to cases in which Jetz is incorrect. Since about 1/3 of the species included in Jetz lacked underlying sequence data, their position in the trees is based upon traditional taxonomy. In some cases, more recent data has shown that there are taxa that are misplaced into the wrong family through this procedure (since taxonomies can be wrong). Was there any attempt to check the "arbitrary" selection to ensure that there was not likely an error in these?

>> We are sorry that the “Arbitrarily selecting” is an over-simplified description. In most cases, we picked species from the type genus, which defines the family and gives the root of the family name, as the representative of that family. For example, we picked up “Sitta europaea” as the representative of the family “Sittidae”. In our dataset, 222 out of 242 representative species belong to the type genera of the focal families.

Furthermore, while the most recent taxonomic revisions are in passerines, only 4 (out of 137) passerine families have representative species not belonging to type genera. Thus, it is unlikely that the representative species were misplaced into wrong families after the recent revision. We added the details of choosing representative species in the Methods section (Line 400-403).

REVIEWERS' COMMENTS:

Reviewer #3 (Remarks to the Author):

The authors have addressed my previous comments, including adding in some additional analyses that I feel strengthened the manuscript. I do not have any additional comments to add.